# FairDICE: Fairness-Driven Offline Multi-Objective Reinforcement Learning

**Woosung Kim**[1][*]    **Jinho Lee**[1][*]    **Jongmin Lee**[2][†]    **Byung-Jun Lee**[1,3][†]

[1]Korea University    [2]Yonsei University    [3]Gauss Labs Inc.
{wsk208,jinho0997,byungjunlee}@korea.ac.kr
jongminlee@yonsei.ac.kr
[*]Equal contribution    [†]Corresponding authors

## Abstract

Multi-objective reinforcement learning (MORL) aims to optimize policies in the presence of conflicting objectives, where linear scalarization is commonly used to reduce vector-valued returns into scalar signals. While effective for certain preferences, this approach cannot capture fairness-oriented goals such as Nash social welfare or max-min fairness, which require nonlinear and non-additive trade-offs. Although several online algorithms have been proposed for specific fairness objectives, a unified approach for optimizing nonlinear welfare criteria in the offline setting—where learning must proceed from a fixed dataset—remains unexplored. In this work, we present FairDICE, the first offline MORL framework that directly optimizes nonlinear welfare objective. FairDICE leverages distribution correction estimation to jointly account for welfare maximization and distributional regularization, enabling stable and sample-efficient learning without requiring explicit preference weights or exhaustive weight search. Across multiple offline benchmarks, FairDICE demonstrates strong fairness-aware performance compared to existing baselines.

## 1   Introduction

Sequential decision-making in real-world domains often requires balancing multiple conflicting objectives, as seen in applications like autonomous driving [1, 2], robotic manipulation [3, 4], and wireless network resource allocation [5]. Multi-objective reinforcement learning (MORL) addresses this challenge by providing a principled framework for learning policies that maximize aggregated returns over conflicting objectives. While standard MORL with linear scalarization focuses on maximizing a weighted sum of objective returns, MORL with nonlinear scalarization, or fair MORL, promotes fair outcomes through concave scalarization objectives, such as Nash social welfare [6].

The nonlinearity of fair MORL presents a major optimization challenge due to its nonlinear scalarization over objective returns and has been widely studied in online settings where agents learn through interaction. Some approaches maximize a lower bound—the expected scalarized return [7, 8]—while others focus solely on max-min fairness or use policy-gradient methods to directly optimize the original objective[9, 10]. However, fair MORL in the offline setting remains unexplored where the agent needs to learn fair policy from a fixed dataset and avoids risky or costly interaction with the environment.

Recent studies have explored offline MORL but primarily focus on linear scalarization, learning policies conditioned on fixed preference weights [11] to perform well along the Pareto front [12, 13]. However, these methods are unsuitable for fair MORL, which aims to maximize its fairness objectives (welfare) without explicitly specifying preferences. To this end, we formulate offline fair MORL

problem to directly optimize the trade-off between welfare and distribution shift regularization required for offline RL. To the best of our knowledge, this is the first work to optimize a welfare-maximizing policy from a fixed dataset.

In this paper, we also show that while MORL with linear scalarization and fair MORL are fundamentally distinct problems with different solution spaces, they can be theoretically connected under offline regularization, sharing the same optimal solution. Building on this insight, we extend the DICE-RL framework, which optimizes the stationary distribution for offline policy learning, to handle the nonlinearity of fair MORL and develop FairDICE, our sample-based offline MORL algorithm. FairDICE effectively finds welfare-optimal policies in both discrete and continuous domains with minimal additional parameters, outperforming preference-conditioned baselines even with exhaustive weight search. In summary, our contributions are threefold:

- A regularized offline MORL formulation that optimizes nonlinear welfare objectives while mitigating distributional shift.
- A theoretical connection between our formulation and linear scalarization under regularization, showing that **FairDICE** implicitly optimizes preference weights for welfare maximization.
- **FairDICE**, a practical, sample-based algorithm for offline welfare optimization.

## 2    Related Work

**Multi-Objective RL and Welfare Objectives**    Linear scalarization is a common approach within MORL that optimizes a weighted sum of returns [14]. Nevertheless, it fails to capture complex trade-offs like fairness or risk sensitivity [15, 16, 17, 18]. This limitation has led to growing interest in nonlinear scalarization, which enables more expressive preference modeling. Recent theoretical work has shown that it is tractable to optimize nonlinear scalarizations under smooth, concave utility functions [19, 10]. Such scalarization functions, including Nash social welfare and Gini indices, are optimized via online interactions [7, 20]. Max-min objectives, another form of fairness-aware scalarization, have been addressed in model-free online settings with entropy regularization [9].

**Offline RL and Offline MORL**    Offline reinforcement learning aims to learn policies from fixed datasets without further environment interaction, avoiding costly or risky exploration. A major challenge in this setting is distribution shift: deviations from the behavior policy can lead to inaccurate value estimates and suboptimal performance. To address this, various strategies have been proposed, including conservative value estimation [21], divergence-regularized optimization [22, 23], and return-conditioned sequence modeling [24]. In the multi-objective setting, most offline approaches assume linear scalarization and require explicit preference conditioning during training or evaluation [25, 26]. However, direct optimization of nonlinear objectives in offline MORL remains largely underexplored.

## 3    Preliminaries

### 3.1    Markov Decision Process (MDP) and Multi-objective RL

A Markov Decision Process (MDP) is defined by the tuple $(\mathcal{S}, \mathcal{A}, T, r, p_0, \gamma)$, where $\mathcal{S}$ and $\mathcal{A}$ are the state and action spaces, $T$ is the transition probability, $r$ is the reward function, $p_0$ is the initial state distribution, and $\gamma \in [0, 1)$ is the discount factor. For a policy $\pi$, the stationary distribution $d_\pi(s, a)$ represents the discounted visitation frequency of state-action pairs and satisfies the Bellman flow constraint:

$$d_\pi(s, a) = \pi(a|s) \left( (1 - \gamma)p_0(s) + \gamma \sum_{\bar{s}, \bar{a}} T(s|\bar{s}, \bar{a})d_\pi(\bar{s}, \bar{a}) \right), \ \forall s, a \tag{1}$$

The expected return of policy $\pi$, defined as the discounted sum of rewards, can be represented as $J(\pi) := \sum_{s,a} d_\pi(s, a)r(s, a)$, and standard RL aims to find $\pi$ that maximizes $J(\pi)$.

In Multi-Objective Reinforcement Learning (MORL), the return is represented as a vector $[J_1(\pi), \dots, J_I(\pi)]$, where each $J_i(\pi)$ is defined under a corresponding reward function $r_i(s, a)$.

MORL methods typically scalarize the return vector to optimize a scalar objective of the form $\sum_i u_i(J_i(\pi))$. Linear scalarization applies fixed weights $w_i$, reducing the problem to single-objective RL with the aggregated reward $r(s,a) = \sum_i w_i r_i(s,a)$. This approach focuses on recovering optimal policies across different weight configurations.

However linear scalarization does not consider fairness across objectives. To address this, we use strictly concave scalarization functions, referred to as *Fair MORL*, which maximize the welfare $\sum_i u_i(J_i(\pi))$, as in Nash Social Welfare with $u_i(x) = \log(x)$. However, this nonlinearity breaks the Bellman recursion, making Temporal Difference (TD) targets difficult to define and thus unsuitable for standard value-based RL methods.

### 3.2 Offline Reinforcement Learning and DICE-RL framework

In offline reinforcement learning (RL), an agent learns a policy from a fixed dataset $D = \{s_i, a_i, r_i, s_i'\}_{i=1}^N$, without further environment interaction, making it well-suited for scenarios where exploration is costly or unsafe. A core challenge in offline RL is distribution shift, where the learned policy deviates from the behavior policy to improve performance. Excessive deviation leads to significant off-policy estimation errors and degrades real-world performance.

To address this, the DICE-RL framework [22] regularizes policy optimization using an $f$-divergence between the stationary distribution of the learned policy $d$ and the empirical distribution $d_D$, solving:

$$\max_{d \geq 0} \sum_{s,a} d(s,a)r(s,a) - \beta \sum_{s,a} d_D(s,a)f\left(\frac{d(s,a)}{d_D(s,a)}\right)$$

$$\text{s.t.} \sum_a d(s,a) = (1-\gamma)p_0(s) + \gamma \sum_{\bar{s},\bar{a}} T(s|\bar{s},\bar{a})d(\bar{s},\bar{a}), \quad \forall s \qquad (2)$$

where the marginalized Bellman flow constraint (2) ensures the optimized stationary distribution $d$ is valid. The hyperparameter $\beta > 0$ controls the trade-off between return maximization and distribution shift regularization. In finite domains, the optimal policy $\pi^*$ is recovered from $d^*$ via $\pi^*(a|s) = \frac{d^*(s,a)}{\sum_a d^*(s,a)} \ \forall s, a$, with policy extraction methods used in continuous domains. Full derivations appear in Appendix A.

## 4 Fair MORL as Convex Optimization

We present a convex optimization formulation for Fair MORL. Before extending it to a practical offline algorithm, we analyze its structure and reveal a connection to the Eisenberg–Gale program [27]. We reformulate the objective to facilitate future sample-based methods and highlight key differences from linear scalarization.

### 4.1 Convex Formulation via stationary distribution

We introduce a convex formulation of Fair MORL, where the objective returns $J_i(\pi)$ are expressed in terms of the stationary distribution. This formulation extends the dual of V-LP [28], the linear programming formulation of RL, by replacing its linear objective with a welfare function, a similar approach explored in [9, 10]. Specifically, we consider the following problem, where each $u_i(x)$ is strictly concave:

$$(\text{P1}): \max_{d \geq 0} \sum_i u_i\left(\sum_{s,a} d(s,a)r_i(s,a)\right)$$

$$\text{s.t.} \ \mathcal{F}_d(s) = 0, \quad \forall s$$

where $\mathcal{F}_d(s) = (1-\gamma)p_0(s) + \gamma \sum_{\bar{s},\bar{a}} T(s|\bar{s},\bar{a})d(\bar{s},\bar{a}) - \sum_a d(s,a)$ denotes the violation degree of the Bellman flow constraint (2).

Uniqueness of the solution to (P1) follows from the strict concavity of the objective and the convex, non-empty feasible set of valid stationary distributions. When $u_i(x) = \log(x) \ \forall i$, the optimization takes a form similar to the Eisenberg–Gale convex program, a classical model for computing fair and

Pareto-efficient market equilibria:

$$\max_{x_{ij} \geq 0} \sum_i \log \left( \sum_j u_{ij} x_{ij} \right) \quad \text{s.t.} \sum_i x_{ij} \leq 1 \, \forall j$$

where $x_{ij}$ denotes the allocation of good $j$ to buyer $i$, and $u_{ij}$ is the utility of buyer $i$. In our setting, this corresponds to allocating actions in a single-state MDP to maximize Nash social welfare, with $i$ representing actions and $j$ representing objectives. With additional constraint (2), (P1) can be viewed as a sequential version of the Eisenberg–Gale program, extending the allocation problem to sequential decision-making.

## 4.2 Reformulation for sample-based optimization

Problem (P1) involves an expectation inside a concave function, which complicates the future derivation of our sample-based optimization method. To address this, we reformulate (P1) as (P2) by introducing slack variables, moving the expectation outside the concave function.

$$\text{(P2):} \max_{d \geq 0, k_i} \sum_i u_i(k_i) \quad \text{s.t.} \sum_{s,a} d(s,a) r_i(s,a) = k_i \, \forall i, \quad \mathcal{F}_d(s) = 0, \, \forall s$$

where $k_i$ is a slack variable representing the expected return for objective $i$. The Lagrangian dual of (P2) introduces Lagrange multipliers $\mu_i$ for the return constraints and $\nu(s)$ for the Bellman flow constraints. The dual formulation is given by:

$$\max_{d \geq 0, k} \min_{\nu, \mu} \sum_i u_i(k_i) + \sum_i \mu_i \left( \sum_{s,a} d(s,a) r_i(s,a) - k_i \right) + \sum_s \nu(s) \mathcal{F}_d(s) \tag{3}$$

In the dual function (3), the Lagrange multiplier $\mu_i$ modulates each objective's return, resembling the role of preference weights in Linear MORL. At optimality, the following relationship holds:

$$\mu_i^* = u_i'(k_i^*) = u_i' \left( \sum_{s,a} d^*(s,a) r_i(s,a) \right)$$

Since $u_i'(x)$ is decreasing due to the strict concavity of $u_i$, $\mu_i$ acts as an implicit preference weight that penalizes large returns. For example, when $u_i(x) = \log(x)$, this yields $\mu_i^*$ that corresponds to the reciprocal of $i$th return, assigning higher weight to objectives with lower returns and promoting a more balanced, fair allocation—consistent with the goal of Nash social welfare and Fair MORL.

However, although $\mu_i$ plays a role similar to preference weights in Linear MORL, fixing $\mu_i^*$ in Linear MORL (P3) leads to a fundamentally different optimization problem from (P2) learning $\mu_i$ as part of the nonlinear welfare objective. (P3) is defined below, and a simple counterexample are provided in Appendix B.

$$\text{(P3):} \max_{d \geq 0} \sum_i \mu_i^* \sum_{s,a} d(s,a) r_i(s,a) \quad \text{s.t.} \, \mathcal{F}_d(s) = 0, \quad \forall s$$

Unlike (P1), which has a unique solution by strict concavity, (P3) may have multiple optimal solutions with different welfare outcomes. Thus, the welfare-maximizing policy of Fair MORL cannot generally be found by simply sweeping over weight vectors in Linear MORL.

## 5 FairDICE: Welfare Optimization for Offline Fair MORL

In this section, we introduce the Regularized Welfare Optimization framework and a corresponding sample-based algorithm for effective welfare optimization in the offline setting. Building on the DICE-RL framework applied to (P2), our method, FairDICE, directly optimizes implicit preference weights to maximize welfare. We further provide theoretical support by recovering an equivalent Regularized Linear MORL formulation.

## 5.1 Regularized Welfare Optimization framework

We formulate our framework by incorporating an $f$-divergence between the optimized stationary distribution $d$ and the empirical data distribution $d_D$ into (P2). The trade-off between the welfare and distributional shift is controlled by a hyperparameter $\beta > 0$, and $f$ is assumed to be strictly convex with $f(1) = 0$. The resulting convex optimization is:

$$\text{(P2-reg): } \max_{d \geq 0, k} \sum_i u_i(k_i) - \beta \sum_{s,a} d_D(s,a) f\left(\frac{d(s,a)}{d_D(s,a)}\right)$$

$$\text{s.t. } \sum_{s,a} d(s,a) r_i(s,a) = k_i \ \forall i, \quad \mathcal{F}_d(s) = 0, \ \forall s$$

Following the DICE-RL framework, we derive a sample-based optimization method from the Lagrangian dual of (P2-reg). We also highlight the challenges of extending (P1) directly to sample-based optimization and show how (P2) circumvents the issues. The Lagrangian is expressed as:

$$\max_{d \geq 0, k} \min_{\nu, \mu} L(\nu, \mu, d, k) := \sum_i \mu_i \left( \sum_{s,a} d(s,a) r_i(s,a) - k_i \right) - \beta \sum_{s,a} d_D(s,a) f\left(\frac{d(s,a)}{d_D(s,a)}\right)$$

$$+ \sum_i u_i(k_i) + \sum_s \nu(s) \mathcal{F}_d(s)$$

We reparameterize the stationary distribution as $d(s,a) = w(s,a) d_D(s,a)$ and express the dual in terms of the importance weights $w$, using the identity $\sum_s \nu(s) \sum_{\bar{s}, \bar{a}} T(s | \bar{s}, \bar{a}) d(\bar{s}, \bar{a}) = \sum_{s,a} d(s,a) \sum_{s'} T(s' | s, a) \nu(s')$. This yields the following optimization problem:

$$\max_{w \geq 0, k} \min_{\nu, \mu} \mathbb{E}_{s \sim p_0}[(1 - \gamma) \nu(s)] + \mathbb{E}_{d_D}\left[ w(s,a) e_{\nu,\mu}(s,a) - \beta f(w(s,a)) \right] - \sum_i (\mu_i k_i + u_i(k_i))$$

where $e_{\nu,\mu}(s,a) = \sum_i \mu_i r_i(s,a) + \gamma \sum_{s'} T(s'|s,a) \nu(s') - \nu(s) \ \forall s, a$.

Applying the Lagrangian dual directly to the regularized (P1) retains expected returns inside the concave functions $u_i(\cdot)$, preventing direct use of importance sampling. Moreover, a naive estimator such as $\sum_{s,a} d_D(s,a) \sum_i u_i(w(s,a) r(s,a))$ introduces bias, violating the validity of importance-weighted estimation.

We further simplify the optimization by reducing parameters. Using strong duality of (P2-reg), we switch the optimization order to $\min_{\nu,\mu} \max_{w,k}$ and derive closed-form solutions for $w$ and $k_i$ from first-order conditions:

$$w^*_{\nu,\mu}(s,a) = \max\left(0, (f')^{-1}\left(\frac{e_{\nu,\mu}(s,a)}{\beta}\right)\right) \ \forall s, a, \quad k^*_{i,\mu} = (u'_i)^{-1}(\mu_i) \ \forall i$$

Substituting the closed-form solutions into the Lagrangian dual yields the final optimization, which defines the loss function of our offline algorithm, FairDICE (Fair MORL via Stationary Distribution Correction):

$$\min_{\nu,\mu} \mathbb{E}_{s \sim p_0}[(1 - \gamma) \nu(s)] + \mathbb{E}_{(s,a) \sim d_D}\left[ \beta f_0^*\left(\frac{e_{\nu,\mu}(s,a)}{\beta}\right) \right] + \sum_i u_i^*(-\mu_i) \tag{4}$$

where $f^*(y) := \max_{x \geq 0} xy - f(x)$ and $u_i^*(y) := \max_x xy + u_i(x)$ are convex conjugate functions. Solving (4) gives the optimal stationary distribution $d^*(s,a) = w^*_{\nu^*, \mu^*}(s,a) d_D(s,a)$. In finite domains, the optimal policy is directly recovered via $\pi^*(a|s) = d^*(s,a) / \sum_a d^*(s,a) \ \forall s, a$.

Compared to its single-objective counterpart, OptiDICE, FairDICE introduces only one additional scalar parameter per objective, incurring minimal overhead and scaling efficiently with the number of objectives. Moreover, FairDICE extends naturally to large and continuous domains by approximating $\nu, \mu$ and $\pi$ with function approximators. In continuous domains, we adopt weighted behavior cloning to extract the policy from the optimal stationary distribution using the following loss:

$$\max_{\pi_\phi} \mathbb{E}_{(s,a) \sim d_D}\left[ w^*_{\nu^*, \mu^*}(s,a) \log \pi_\phi(a|s) \right]$$

Further details in experimental settings and algorithmic descriptions for continuous domains are provided in Appendix F.

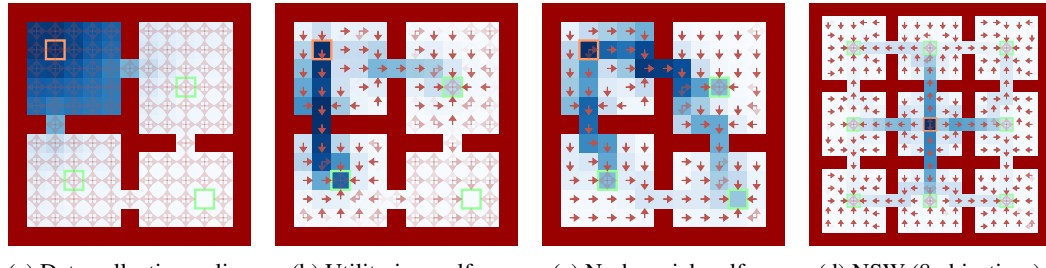

| (a) Data collection policy | (b) Utilitarian welfare | (c) Nash social welfare | (d) NSW (8 objectives) |

Figure 1: Visualization of FairDICE policies in MO-Four-Rooms: (a) Uniformly random policy for data collection, (b) FairDICE policy maximizing Utilitarian welfare (sum of returns), (c) FairDICE policy maximizing Nash social welfare (NSW). (d) FairDICE maximizing NSW in a domain with eight objectives. Red arrows indicate the policy, and the blue heatmap shows state visitation.

## 5.2 Equivalence to Regularized Linear MORL

Previously, we showed that unregularized linear and fair MORL converge to different solutions. We now demonstrate that their regularized forms converge to the same unique solution. This equivalence supports that FairDICE implicitly optimizes preference weights corresponding to those in linear MORL to directly maximize target welfare.

To formalize this, we introduce (P3-reg), an extension of DICE-RL framework to Linear MORL. While any preference weight can be used, we set it to the optimal dual variable $\mu^*$ from (4) for analysis:

$$\text{(P3-reg): } \max_{d \geq 0} \sum_{s,a} d(s,a) \sum_i \mu_i^* r_i(s,a) - \beta \sum_{s,a} d_D(s,a) f\left(\frac{d(s,a)}{d_D(s,a)}\right)$$
$$\text{s.t. } \mathcal{F}_d(s) = 0, \quad \forall s$$

**Proposition 1** (Equivalence between the regularized problems). *Let $\mu_i^*$ be the optimal multipliers obtained from (4). Then, the optimal solutions of FairDICE and (P3-reg) with $\mu_i^*$ yield the same unique optimal policy. (Proof in Appendix C)*

This equivalence suggests that FairDICE reduces to an offline Linear MORL algorithm when its $\mu$ is fixed to the preference weights. We refer to this special case as FairDICE-fixed and leverage it to optimize utilitarian welfare—the sum of objective returns—by setting all $\mu$s equal.

The equivalence also implies that sweeping over preference weights in regularized Linear MORL can recover the optimal policy found by FairDICE. However, this requires training policies across a wide range of weights and selecting the one that achieves the highest welfare, which becomes impractical as the number of objectives increases. In contrast, FairDICE effectively and efficiently optimizes implicit preference weights, directly producing an offline MORL policy that maximizes welfare.

## 6 Empirical Behaviors of FairDICE

In this section, we empirically validate our theoretical insights using a multi-objective adaptation of the classic Four-Room environment [22, 29] and Random MDP [22, 30] as a toy example. The visualization shows how FairDICE effectively optimizes the trade-off between welfare and distribution shift (Section 5.1) and aligns with Regularized Linear MORL while optimizing its implicit preference weight for offline welfare optimization (Section 5.2).

In the experiments, we use $\alpha$-fairness to aggregate objectives, a generalized social welfare function that balances total return and fairness. The trade-off is controlled by the parameter $\alpha$: as $\alpha \to 0$, it approximates Utilitarian welfare, the sum of returns; at $\alpha = 1$, it recovers Nash social welfare; and as $\alpha \to \infty$, it approaches the max-min fairness. The scalarization function is defined as:

$$u_i(x) = \begin{cases} (1-\alpha)^{-1} x^{1-\alpha} & (\alpha \neq 1) \\ \log(x) & (\alpha = 1) \end{cases} \quad \forall i$$

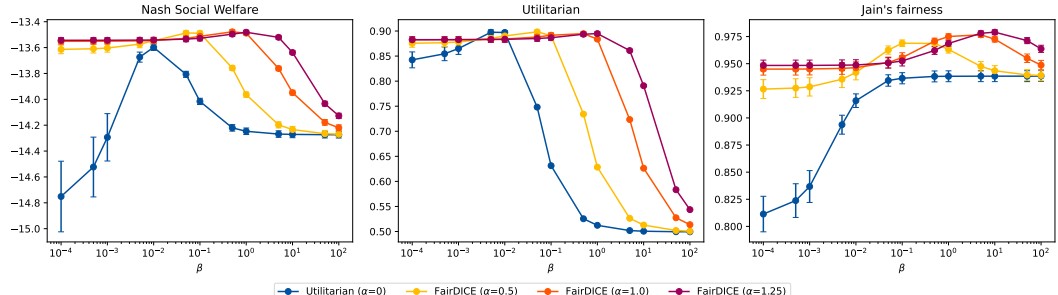

Figure 2: Policy performance on Random MOMDP domain across different $\alpha$ and $\beta$ values, evaluated on Nash social welfare, Utilitarian welfare, and Jain's fairness index. Results are averaged over 1000 seeds, and reported with 95 % confidence intervals.

We evaluate the resulting policies using three metrics: **Utilitarian welfare** $\sum_{i=1}^n R_i$, **Jain's Fairness Index** $\left(\sum_{i=1}^n R_i\right)^2 / \left(n \sum_{i=1}^n R_i^2\right)$ and **Nash social welfare (NSW)** $\sum_{i=1}^n \log(R_i)$. Utilitarian welfare measures the total return, Jain's Fairness Index evaluates fairness across objectives, and Nash social welfare captures a trade-off between efficiency and fairness. Details of the environments and experiments introduced in this section are provided in Appendix D.

## 6.1 MO-Four-Room Experiment

We extend the Four-Room domain to a MORL setting, referred to as MO-Four-Room, by introducing three distinct goals, each associated with a separate objective. As shown in Figure 1, the agent starts from the initial state (orange) and moves toward the goal states (green). Upon reaching a goal, it receives a one-hot reward: $[1, 0, 0]$, $[0, 1, 0]$, or $[0, 0, 1]$. To simulate offline RL, we construct a dataset of 300 trajectories collected from a uniformly random behavior policy.

Even when data is collected under an unfair, suboptimal policy, FairDICE successfully learns offline MORL policies that maximize target welfare. Figure 1 shows how varying $\alpha$-fairness objectives shape FairDICE's behavior. The utilitarian objective favors the nearest goal as it results in the best return, achieving the highest sum of returns despite being unfair. However,

Table 1: MO-Four-Room Performance Table

|      | Behavior | Utilitarian | FairDICE-NSW |
|------|----------|-------------|--------------|
| NSW  | -16.10   | -17.33      | **-10.77**   |
| Util | 0.024    | **0.086**   | 0.082        |
| Jain | 0.710    | 0.500       | **0.996**    |

FairDICE maximizing Nash social welfare (FairDICE-NSW), encourages balanced goal visitation, even when the environment is expanded to include eight objectives, as shown in Figure 1d.

## 6.2 Effective Optimization over Two Distinct Trade-offs

The Regularized Welfare Optimization framework balances two trade-offs controlled by $\alpha$ and $\beta$: one between objective returns under $\alpha$-fairness and the other between welfare and distributional shift. To empirically demonstrate how these trade-offs evolve with varying parameters and generalize across different MOMDP environments, we extend the Random MDP to a MORL setting, called Random MOMDP, following a similar approach used for the Four-Room domain.

Figure 2 illustrates how the three performance metrics vary under these trade-offs. Increasing $\alpha$ shifts the objective toward max-min fairness, improving Jain's fairness index, while decreasing $\alpha$ prioritizes total return, enhancing utilitarian performance. Higher $\beta$ values constrain the learned policies to remain closer to the data collection policy across all metrics. In contrast, lower $\beta$ values reduce regularization, allowing each objective to more effectively pursue its own $\alpha$-fairness. However, if $\beta$ is excessively low, the resulting distribution shift can degrade practical performance.

A surprising finding is that while FairDICE achieves the highest utilitarian performance when maximizing utilitarian welfare ($\alpha = 0$), higher $\alpha$ values also sustain strong utilitarian performance across a wide range of $\beta$. This stems from the concavity of the $\alpha$-fairness objective, where returns contribute less to overall welfare as they grow. Consequently, the incentive for pure return maximization is tempered, implicitly regularizing against distributional shift.

## 6.3 Welfare Optimization via Implicit preference weight

In Section 5.2, we established the equivalence between regularized linear and fair MORL. We validate this by perturbing the optimal $\mu^*$ obtained by FairDICE-NSW in the Random MOMDP experiment and apply it to FairDICE-fixed, as reported in Figure 3. Gaussian noise with standard deviation $\sigma$ is generated and applied to each dimension of $\mu^*$ by scaling it as $(1+\text{noise})$. FairDICE-fixed achieves the highest NSW without perturbation, and the NSW decreases as the preference weights deviate from the optimal value. This indicates that our regularized welfare optimization framework shares the same optimal solution with regualrized MORL with linear scalarization and optimizes its implicit preference that maximize NSW.

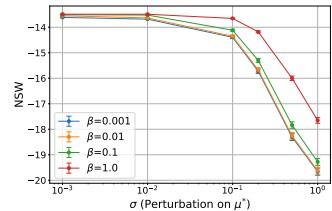

Figure 3: FairDICE-fixed with perturbed $\mu^*$

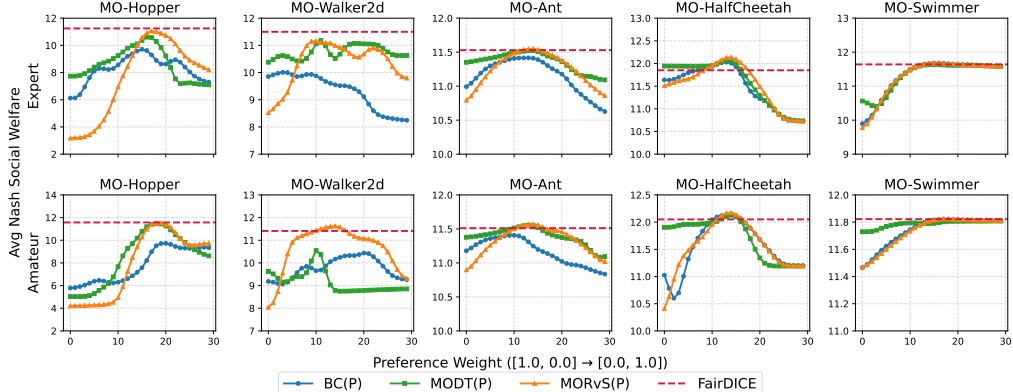

Figure 4: Nash social welfare scores for five two-objective tasks, evaluated across 30 linearly spaced preference weights. Each curve shows the average NSW over 5 seeds and 10 evaluation episodes per seed. Red line indicates the average NSW performance of FairDICE.

# 7 Welfare Maximization in Continuous Domains

**Environments** We evaluate our method on the D4MORL benchmark [31], a standard MORL benchmark in continuous control domains. D4MORL builds upon the D4RL benchmark [32] by decomposing the original MuJoCo rewards into multiple objectives, such as speed, height and energy efficiency. The dataset for each domain consists of two types of data, collected using either expert or stochastically perturbed (amateur) behavioral policies, and is annotated with preference vectors. Our main experiments include five two-objective tasks (e.g., MO-Hopper, MO-Walker2d, MO-HalfCheetah, MO-Ant, MO-Swimmer) and one three-objective task (MO-Hopper-3obj). Further details on the environments and experiments are reported in Appendix E.

**Baselines** While FairDICE seeks a single offline RL policy that maximizes welfare, no existing algorithm directly optimizes this objective in offline MORL. Therefore, we compare against three offline MORL approaches with linear scalarization, additionally searching for preference weights that maximize NSW by uniformly discretizing the simplex and evaluating NSW at each point. Specifically, we adopt three baselines that learn preference-conditioned policies [31].

- **BC(P)** performs behavioral cloning by conditioning on the state and a preference vector to imitate observed actions.
- **MODT(P)** extends Decision Transformer by modeling trajectories as sequences of (state, action, return-to-go) tokens concatenated with a preference vector, using a transformer to predict actions.
- **MORvS(P)** simplifies this setup using a feedforward model that takes the current state and preference-weighted return-to-go as input, enabling more efficient training.

While the baseline includes approaches that do not concatenate the state and linear preference ratio, we do not evaluate them as they generally underperform compared to their preference-conditioned

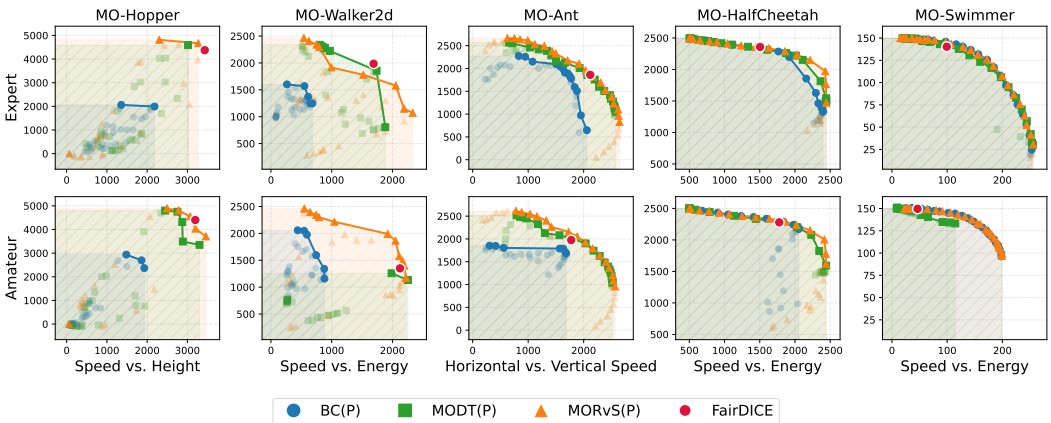

Figure 5: Raw return evaluations on five two-objective MuJoCo tasks from D4MORL. Each point represents policy performance under a specific preference weight; Pareto frontiers and dominated regions are shown.

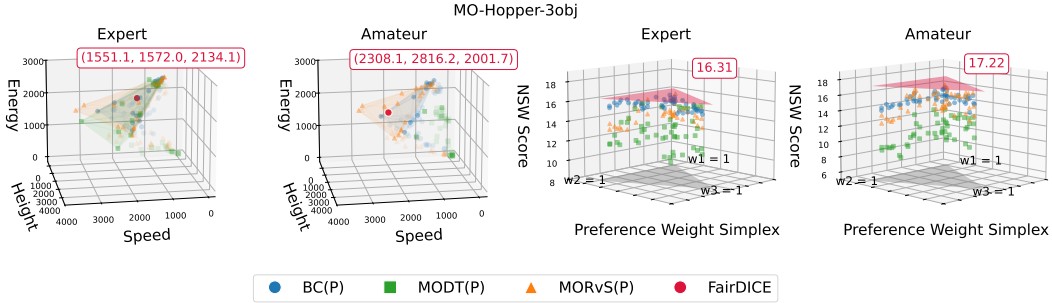

Figure 6: Raw returns and Nash social welfare evaluations on MO-Hopper-3obj with three objectives: speed, height, and energy. 50 preference weights are sampled uniformly from the 3D simplex. Red plane indicates average NSW performance of FairDICE.

counterparts. In contrast, our method does not assume behavior policy preference weights but directly optimizes them to maximize welfare, making it applicable even to datasets without this information.

**Offline Fair MORL Performance** Figure 4 shows that FairDICE achieves competitive or superior NSW performance across all two-objective D4MORL tasks, compared to the best results obtained by extensively searching over preference weights in existing methods. Since NSW summarizes multiple objectives into a single scalar, it does not fully capture how well each objective is optimized. To better illustrate FairDICE's effectiveness across all objectives, we also position its raw returns relative to the Pareto frontier formed by preference-conditioned baselines. Figure 5 shows that the FairDICE solution lies on the Pareto frontier, highlighting the strong practical performance of our approach.

In the MO-Hopper-3obj task with three objectives, the preference weight space expands substantially, making it increasingly difficult to find weights that maximize NSW using MORL with linear scalarization. However, Figure 6 illustrates that FairDICE identifies a welfare-maximizing policy without requiring explicit preference conditioning. Notably, while optimizing for NSW, the resulting policy also achieves raw returns that lie close to or even surpass the Pareto frontier formed by preference-conditioned baselines, indicating that FairDICE achieves strong efficiency in addition to fairness. Moreover, a single additional scalar parameter is sufficient to handle the increased number of objectives, highlighting the scalability and practical efficiency of FairDICE, particularly in high-dimensional preference spaces.

# 8 Conclusion

In this paper, we introduce a novel regularized welfare optimization framework for maximizing welfare in offline MORL, enabling fair outcomes across objectives using fixed datasets—a setting

not addressed by prior work. We establish a theoretical connection between regularized MORL with linear scalarization, showing that our framework implicitly learns preference weights that maximize welfare. Building on this, we extend the DICE RL framework to derive our sample based algorithm, **FairDICE**, that overcomes the optimization challenge caused by the nonlinearity of the objective. Empirically, FairDICE achieves strong fairness aware performance across both discrete and continuous domains with a fixed dataset, effectively balancing trade-offs between objectives and between welfare and distributional shift.

**Limitation** While our method effectively handles MORL with strictly concave scalarization functions, it does not cover all forms of nonlinear scalarization. Our formulation assumes convexity, and thus the DICE-based approach relying on the Lagrangian dual does not apply to non-convex scalarization in MORL. Additionally, although concave scalarization helps mitigate sensitivity to distribution shift, as an offline RL method, the final performance still depends on the choice of hyperparameters controlling distribution shift and the quality of the dataset.

## 9  Acknowledgements

This work was partly supported by Institute of Information & Communications Technology Planning & Evaluation (IITP) grant funded by the Korea government (MSIT) (No. RS-2022-II220311, Development of Goal-Oriented Reinforcement Learning Techniques for Contact-Rich Robotic Manipulation of Everyday Objects, No. RS-2024-00457882, AI Research Hub Project, No. RS-2019-II190079, Artificial Intelligence Graduate School Program (Korea University), and No. RS-2025-25410841, Beyond the Turing Test: Human-Level Game-Playing Agents with Generalization and Adaptation), the IITP (Institute of Information & Communications Technology Planning & Evaluation)-ITRC (Information Technology Research Center) grant funded by the Korea government (Ministry of Science and ICT) (IITP-2025-RS-2024-00436857), the NRF (RS-2024-00451162) funded by the Ministry of Science and ICT, Korea, BK21 Four project of the National Research Foundation of Korea, the National Research Foundation of Korea (NRF) grant funded by the Korea government (MSIT) (RS-2025-00560367, RS-2025-24803384), the IITP under the Artificial Intelligence Star Fellowship support program to nurture the best talents (IITP-2025-RS-2025-02304828) grant funded by the Korea government (MSIT), and KOREA HYDRO & NUCLEAR POWER CO., LTD (No. 2024-Tech-09). This work was also supported by the Institute of Information & Communications Technology Planning & Evaluation (IITP) grant (RS-2020-II201361, Artificial Intelligence Graduate School Program (Yonsei University) and the Yonsei University Research Fund of 2025-22-0158.

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

## A DICE-RL framework

In this section, we introduce the DICE-RL framework along with its corresponding offline single-objective reinforcement learning algorithm, OptiDICE, as proposed in [22]. Our algorithm can be viewed as a multi-objective extension of OptiDICE. The DICE-RL framework is an offline RL framework where return maximization is regularized with an $f$-divergence between the stationary distribution of the learned policy $d$ and the empirical distribution $d_D$, solving:

$$\max_{d \geq 0} \sum_{s,a} d(s,a)r(s,a) - \beta \sum_{s,a} d_D(s,a)f\left(\frac{d(s,a)}{d_D(s,a)}\right)$$

$$\text{s.t. } \sum_a d(s,a) = (1-\gamma)p_0(s) + \gamma \sum_{\bar{s},\bar{a}} T(s|\bar{s},\bar{a})d(\bar{s},\bar{a}), \quad \forall s$$

where the Bellman flow constraints ensure that the optimal $d^*(s,a)$ constitutes a valid stationary distribution. Lagrangian dual of the convex optimization problem is given by,

$$\max_{d \geq 0} \min_\nu \sum_{s,a} d(s,a)r(s,a) + \sum_s \nu(s)\mathcal{F}_d(s) - \beta \sum_{s,a} d_D(s,a)f\left(\frac{d(s,a)}{d_D(s,a)}\right)$$

where $\mathcal{F}_d(s) = (1-\gamma)p_0(s) + \gamma \sum_{\bar{s},\bar{a}} T(s|\bar{s},\bar{a})d(\bar{s},\bar{a}) - \sum_a d(s,a)$. We reparameterize the stationary distribution as $d(s,a) = w(s,a)d_D(s,a)$ and express the dual in terms of the importance weights $w$, using the identity $\sum_s T(s|\bar{s},\bar{a})\nu(s)\sum_{\bar{s},\bar{a}} d(\bar{s},\bar{a}) = \sum_{s,a} d(s,a)\sum_{s'} T(s'|s,a)\nu(s')$. This yields the following optimization problem:

$$\max_{w \geq 0} \min_\nu L(w,\nu) := \mathbb{E}_{s \sim p_0}[(1-\gamma)\nu(s)] + \mathbb{E}_{(s,a) \sim d_D}[w(s,a)e_\nu(s,a) - \beta f(w(s,a))]$$

where $e_\nu(s,a) = r(s,a) + \gamma \sum_{s'} T(s'|s,a)\nu(s') - \nu(s)\ \forall s,a$.

Strong duality holds by Slater's condition, since the problem is convex and a valid stationary distribution exists in the strictly feasible set. Therefore, the order of optimization can be swapped to $\min_\nu \max_{w \geq 0}$. Optimal $w^*(s,a)$ is computed from the first-order condition given by:

$$\frac{\partial L(w,\nu)}{\partial w(s,a)} = \mathbb{E}_{(s,a) \sim d_D}[e_\nu(s,a) - \beta f'(w(s,a))] = 0$$

where $w_\nu^*(s,a) = \max\left(0, (f')^{-1}\left(\frac{e_\nu(s,a)}{\beta}\right)\right)$. By plugging $w^*(s,a)$ in $L(w,\nu)$ results in the following $\nu$ loss of OptiDICE.

$$\min_\nu \mathbb{E}_{s \sim p_0}[(1-\gamma)\nu(s)] + \mathbb{E}_{(s,a) \sim d_D}\left[\beta f_0^*\left(\frac{e_\nu(s,a)}{\beta}\right)\right]$$

where $f_0^*(y) := \max_{x \geq 0} xy - f(x)$.

After minimizing over $\nu$, the optimal stationary distribution is given by $d^*(s,a) = w_{\nu^*}^*(s,a)d_D(s,a)$. This distribution can then be used to recover the optimal policy that induces $d^*(s,a)$ via $\pi^*(a|s) = \frac{d^*(s,a)}{\sum_a d^*(s,a)}$ for all $s,a$, or through a policy extraction method such as weighted behavior cloning.

## B Counterexample

In this section, we present a counterexample demonstrating that applying the optimal implicit weights $\mu^*$ from Fair MORL (P2) to Linear MORL (P3) does not recover the optimal policy of Fair MORL (P2). Consider an MDP with a single state $s$ and a terminal state reached immediately after taking an action. There are two available actions, $a \in \{a_1, a_2\}$, with corresponding stationary distributions $d(s,a_1)$ and $d(s,a_2)$. Each action yields a reward vector consisting of rewards for objectives $A$ and $B$: $\mathbf{r}(s,a_1) = [r_A(s,a_1), r_B(s,a_1)] = [1, 4]$ and $\mathbf{r}(s,a_2) = [r_A(s,a_2), r_B(s,a_2)] = [3, 1]$.

Using this setup, we construct the corresponding (P1) optimization problem, while assuming Nash social welfare ($u_i = \log(x)\ \forall i$):

$$\text{(P1): } \max_{d \geq 0} \log(d(s,a_1) + 3d(s,a_2)) + \log(4d(s,a_1) + d(s,a_2))$$

$$\text{s.t. } d(s,a_1) + d(s,a_2) = 1$$

where $\log(x)$ is applied to the returns of each objective. An equivalent optimization (P2) is given by:

$$\text{(P2):} \max_{d \geq 0, k} \log(k_1) + \log(k_2)$$

$$\text{s.t. } d(s, a_1) + d(s, a_2) = 1$$
$$d(s, a_1) + 3d(s, a_2) = k_1$$
$$4d(s, a_1) + d(s, a_2) = k_2$$

Its Lagrangian dual is given as,

$$\max_{d \geq 0, k} \min_{\mu, \nu} \log(k_1) + \log(k_2)) + \nu(d(s, a_1) + d(s, a_2) - 1)$$
$$+ \mu_1(d(s, a_1) + 3d(s, a_2) - k_1) + \mu_2(4d(s, a_1) + d(s, a_2) - k_2)$$

By solving the first-order conditions, the optimal Lagrange multipliers are $\mu_1^* \approx 0.5455$ and $\mu_2^* \approx 0.3636$, resulting in the optimal stationary distribution $[d^*(s, a_1), d^*(s, a_2)] = [0.5834, 0.4166]$. The optimal policy of Fair MORL (P2) is a stochastic policy that prefers action $a_1$ while still assigning probability to $a_2$, resulting in objective returns $k^* = [1.8332, 2.7500]$. While $\mu$ is applied to the objective returns in a manner analogous to Linear MORL, we demonstrate that the two formulations are fundamentally distinct by applying the implicit preference weight $\mu^*$ to (P3):

$$\text{(P3):} \max_{d \geq 0} (1 \cdot \mu_1^* + 4 \cdot \mu_2^*)d(s, a_1) + (3 \cdot \mu_1^* + 1 \cdot \mu_2^*)d(s, a_2)$$

$$\text{s.t. } d(s, a_1) + d(s, a_2) = 1$$

In this case, $1 \cdot \mu_1^* + 4 \cdot \mu_2^* = 3 \cdot \mu_1^* + 1 \cdot \mu_2^* = 2.0$, indicating that all policies are the all optimal policies of (P3). As a result, any policy is optimal under Linear MORL with fixed weights $\mu$. This demonstrates that (P2) and (P3) are distinct optimization problems.

## C    Proof of Proposition 1

In this section, we provide a proof of Proposition 1, showing that regularized Linear MORL (P3-reg), when using preference weights equal to the optimal dual variable $\mu^*$ from regularized Fair MORL (P2-reg), converges to the same solution as (P2-reg). To facilitate the explanation, we begin by rewriting the (P3-reg) formulation:

$$\text{(P3-reg):} \max_{d \geq 0} \sum_{s,a} d(s, a) \sum_i \mu_i^* r_i(s, a) - \beta \sum_{s,a} d_D(s, a) f\left(\frac{d(s, a)}{d_D(s, a)}\right) \quad \text{s.t. (2)}$$

The Lagrangian duals of (P2-reg) and (P3-reg) are given by:

$$\max_{d \geq 0, k} \min_{\nu, \mu} L_{\text{P2-reg}}(\nu, \mu, d, k) := \sum_i \mu_i \left( \sum_{s,a} d(s, a) r_i(s, a) - k_i \right) - \beta \sum_{s,a} d_D(s, a) f\left(\frac{d(s, a)}{d_D(s, a)}\right)$$
$$+ \sum_i u_i(k_i) + \sum_s \nu(s) \mathcal{F}_d(s)$$

$$\max_{d \geq 0} \min_{\nu} L_{\text{P3-reg}}(\nu, d) := \sum_{s,a} d(s, a) \sum_i \mu_i^* r_i(s, a) - \beta \sum_{s,a} d_D(s, a) f\left(\frac{d(s, a)}{d_D(s, a)}\right)$$
$$+ \sum_s \nu(s) \mathcal{F}_d(s)$$

Assuming the optimal $\mu^*$ from (P2-reg) is given, we compute the gradient of each Lagrangian with respect to $\nu(s)$ and $d(s, a)$, and show that both share the same gradient at their optimal solutions.

$$\frac{\partial L_{\text{P2-reg}}}{\partial \nu(s)} = \frac{\partial L_{\text{P3-reg}}}{\partial \nu(s)} = \mathcal{F}_{d^*}(s)$$

$$\frac{\partial L_{\text{P2-reg}}}{\partial d(s, a)} = \frac{\partial L_{\text{P3-reg}}}{\partial d(s, a)} = \sum_i \mu_i^* r_i(s, a) - \beta \sum_{s,a} f'\left(\frac{d^*(s, a)}{d_D(s, a)}\right) + \sum_{s'} \gamma T(s'|s, a)\nu^*(s') - \nu^*(s)$$

While (P3) is a linear program, (P3-reg) becomes a convex optimization problem due to regularization with a strictly convex function $f$, making the KKT conditions applicable. From the stationarity conditions, (P3-reg) with $\mu^*$ converges to the same optimal solution as (P2-reg).

## C.1 Empirical evidence

We adapt the counterexample from Appendix B to demonstrate that, under offline regularization, Linear MORL and Fair MORL can share the same optimal solution. In the offline setting, we additionally assume a fixed data distribution over actions given by $[d_D(s, a_1), d_D(s, a_2)] = [0.7, 0.3]$. The (P2-reg) formulation of the counterexample is given by:

$$\text{(P2-reg): } \max_{d \geq 0, k} \ \log(k_1) + \log(k_2) - \sum_a d_D(s, a) f\left(\frac{d(s, a)}{d_D(s, a)}\right)$$

$$\text{s.t. } d(s, a_1) + d(s, a_2) = 1$$
$$d(s, a_1) + 3d(s, a_2) = k_1$$
$$4d(s, a_1) + d(s, a_2) = k_2$$

where we adopt $\chi^2$-divergence $f(x) = \frac{1}{2}(x-1)^2$. The optimal Lagrange multipliers are $\mu_1^* \approx 0.5959$ and $\mu_2^* \approx 0.3352$, resulting in the optimal stationary distribution $[d^*(s, a_1), d^*(s, a_2)] = [0.6609, 0.3390]$. This indicates that as $a_1$ is more common in the data distribution than $a_2$, offline RL policy of (P2-reg) favors $a_1$ compared to the unregularized case. This aligns with the goal of offline reinforcement learning where the optimized stationary distribution should not deviate excessively from the dataset distribution. We apply $\mu^*$ to (P3-reg).

$$\text{(P3-reg): } \max_{d \geq 0} \ (1 \cdot \mu_1^* + 4 \cdot \mu_2^*)d(s, a_1) + (3 \cdot \mu_1^* + 1 \cdot \mu_2^*)d(s, a_2) - \sum_a d_D(s, a) f\left(\frac{d(s, a)}{d_D(s, a)}\right)$$

$$\text{s.t. } d(s, a_1) + d(s, a_2) = 1$$

where $1 \cdot \mu_1^* + 4 \cdot \mu_2^* = 1.936$ and $3 \cdot \mu_1^* + 1 \cdot \mu_2^* = 2.123$.

As (P3-reg) is a convex optimization problem with strictly concave objective, the uniqueness of the optimal solution is guaranteed. The optimal stationary distribution of (P3-reg) is equivalent to that of (P2-reg) as $[d^*(s, a_1), d^*(s, a_2)] = [0.6609, 0.3390]$. This establishes the connection between regularized Linear MORL and Fair MORL theoretically and empirically.

# D  Finite Domain Experiment Setting

In this section, we provide a detailed explanation of the experimental setup used in Section 6. We begin by describing how the classic Four-Room environment [22, 29] and Random MDP [22, 30] are adapted for the offline multi-objective reinforcement learning (MORL) setting. We present additional visualizations of the MO-Four-Room results to further support the findings in Section 6.

## D.1  Environment detail

**MO-Four-Room**  In MO-Four-Room domain, three distinct goals, each associated with a separate objective. The agent starts from the initial state (orange) and navigates toward one of the goal states (green). Upon reaching a goal, it receives a one-hot reward vector: $[1, 0, 0]$ for the lower-left room, $[0, 1, 0]$ for the upper-right room, and $[0, 0, 1]$ for the lower-right room. The agent selects one of four actions, $\{\text{left}, \text{right}, \text{up}, \text{down}\}$; however, the environment is stochastic, and with a probability of 0.1, the agent transitions in a different direction than the one intended. To simulate offline RL, a dataset of 300 trajectories are collected from a uniformly random behavior policy. The experiments are conducted with $\alpha \in \{0.0, 1.0\}$, where $\alpha = 0.0$ corresponds to a policy that maximizes Utilitarian welfare, and $\alpha = 1.0$ corresponds to one that maximizes Nash social welfare. The regularization coefficient is fixed at $\beta = 0.01$ and $f(x) = 0.5(x-1)^2$.

**Random MOMDP**  In the Random MOMDP domain, a multi-objective Markov decision process is generated with $|\mathcal{S}| = 50$, $|\mathcal{A}| = 4$, and discount factor $\gamma = 0.95$. For each state-action pair, the next-state transitions are defined over four possible next states, with transition probabilities sampled from a Dirichlet distribution, $\text{Dir}(1, 1, 1, 1)$. Among the 49 states excluding the fixed initial state, three are randomly selected as goal states, and each is assigned a distinct one-hot reward vector in the same manner as in the MO-Four-Room environment. To simulate the offline RL setting, a dataset of 100 trajectories is collected using a behavior policy with an optimality level of 0.5. Here, optimality is defined as the normalized performance relative to a uniformly random policy $\pi_{\text{unif}}$ (optimality = 0.0) and an optimal policy $\pi^*$ (optimality = 1.0). This

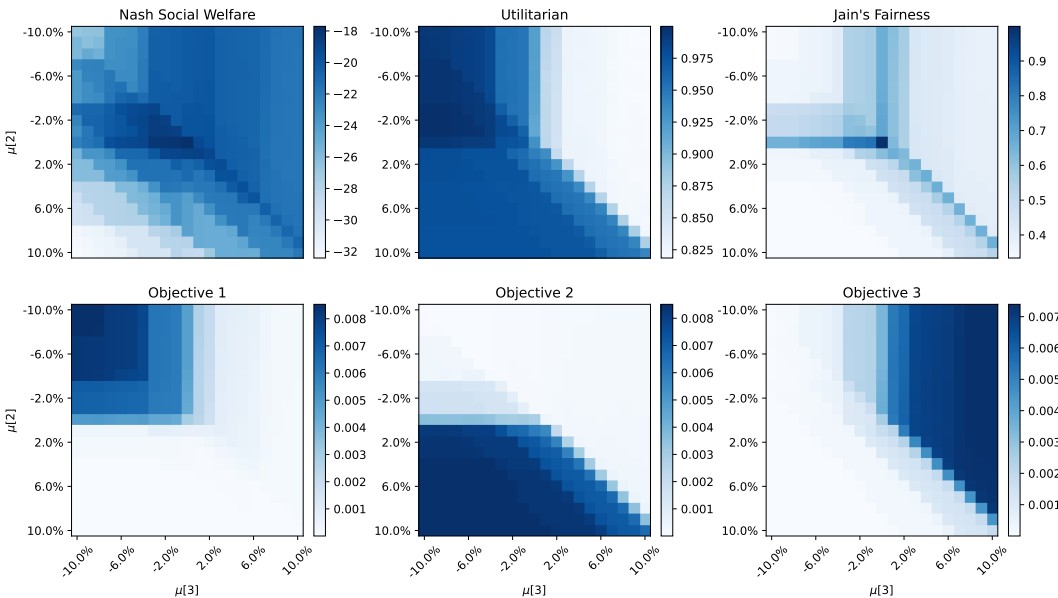

Figure 7: Visualization of FairDICE-fixed performance with different $\mu$ in MO-Four-Room. The center point is the optimal $\mu^*$ obtained by FairDICE. The x-axis and y-axis represent the degrees of perturbation applied to $\mu_2$ and $\mu_3$ from their optimal values.

implies that the behavior policy achieves performance halfway between that of the optimal and random policies. The experiments are repeated over 1000 seeds, with $\alpha \in \{0.0, 0.5, 1.0, 1.25\}$ and $\beta \in \{0.0001, 0.0005, 0.001, 0.005, 0.01, 0.05, 0.1, 0.5, 1.0, 5.0, 10.0, 50.0, 100.0\}$, to provide a comprehensive analysis of the trade-off between welfare maximization and distributional shift.

### D.2 Additional visulaization: MO-Four-Room

Figure 1 illustrates that FairDICE maximizes the welfare objective, resulting in behavior that is distinct from conventional MORL approaches. In this subsection, we extend the experiment from Section 6.2 to further visualize two key insights proposed in our paper: (1) $\mu$ corresponds to the preference weights used in regularized Linear MORL, and (2) these weights are implicitly optimized to maximize the welfare objective—any deviation from the optimal $\mu$ leads to a reduction in overall welfare. In Section 6.2, all preference weights were perturbed within Random MOMDP setting. While this effectively demonstrated that FairDICE selects the welfare-maximizing weights, it is not easy to visualize the consequence of deviation.

Given the optimal preference weight of FairDICE within MO-Four-Room, $\mu^* = [\mu_1, \mu_2, \mu_3]$, we perturb $\mu_2$ and $\mu_3$ while keeping $\mu_1$ fixed. The perturbed weights are applied to FairDICE-fixed to obtain the corresponding optimal policy, whose performance is then evaluated as shown above.

The center point corresponds to the optimal solution of FairDICE, achieving the highest Nash social welfare. Increasing $\mu_2$ boosts return on objective 2, while increasing $\mu_3$ improves return on objective 3. Return on objective 1 increases when both $\mu_2$ and $\mu_3$ decrease. The point that maximizes Utilitarian welfare shifts toward prioritizing objective 1, but this comes at the cost of reduced Nash social welfare and lower Jain's fairness index. These results highlight that FairDICE implicitly optimizes preference weights to maximize welfare, enabling fair behavior in the offline MORL setting.

## E  D4MORL Benchmark

To evaluate the efficacy of offline multi-objective reinforcement learning (MORL) algorithms, we utilize the D4MORL benchmark (with MIT License) introduced by [31]. D4MORL is the large-scale benchmark designed for offline MORL and includes high-dimensional continuous control environments derived from MuJoCo. We adopted settings and configurations of D4MORL without additional modifications.

## E.1 Environments and Objectives

D4MORL comprises six environments: MO-Ant, MO-HalfCheetah, MO-Hopper, MO-Swimmer, and MO-Walker2d, each with two conflicting objectives (e.g., speed vs. energy efficiency), and MO-Hopper-3obj, which includes three objectives—making it a more challenging benchmark. Each environment is defined by multiple, often conflicting, objectives such as forward velocity, jumping stability, or energy consumption. These objectives induce trade-offs, thereby enabling the study of Pareto-optimal policy learning in continuous control settings.

We summarize the objectives for each environment in Table 2, including their physical interpretations and reward formulations. The rewards are computed based on physical quantities such as displacement, height, and control cost. Most environments include a survival bonus term $r_s$ and penalize excessive actions via an action cost term $r_a = \sum_k a_k^2$. The time delta $\Delta t$ determines the resolution of velocity and height-based rewards and is environment-specific.

**Reward Terms and Environment-Specific Constants.** The reward functions in Table 2 include shared terms whose values vary across environments:

- **Action penalty** ($r_a$): Typically defined as the squared sum of action magnitudes, i.e., $r_a = \sum_k a_k^2$. However, different environments apply distinct scaling factors: $r_a = 0.5 \sum_k a_k^2$ in MO-Ant; $r_a = 2 \times 10^{-4} \sum_k a_k^2$ in MO-Hopper.
- **Survival bonus** ($r_s$): A constant reward to encourage survival, set to $r_s = 1.0$ in all environments except MO-Swimmer, where it is omitted.
- **Time delta** ($\Delta t$): Represents the duration between timesteps used in computing velocities or other dynamic terms. Its value is $\Delta t = 0.05$ in MO-Ant, MO-HalfCheetah, and MO-Swimmer; $\Delta t = 0.01$ in MO-Hopper and MO-Hopper-3obj; and $\Delta t = 0.008$ in MO-Walker2d.
- **Initial height** ($h_{\text{init}}$): In MO-Hopper and MO-Hopper-3obj, vertical jump rewards are defined relative to a fixed starting height of $h_{\text{init}} = 1.25$.

These constants follow the environment configurations detailed in Appendix A of [31].

Table 2: Objectives in D4MORL Environments

| Environment | Objective Name | Reward Description |
|---|---|---|
| MO-Ant | $r_{vx}$: velocity in $x$ direction 
 $r_{vy}$: velocity in $y$ direction | $r_{vx} = \frac{x_t - x_{t-1}}{\Delta t} + r_s - r_a$ 
 $r_{vy} = \frac{y_t - y_{t-1}}{\Delta t} + r_s - r_a$ |
| MO-HalfCheetah | $r_v$: forward speed 
 $r_e$: energy efficiency | $r_v = \min(4.0, \frac{x_t - x_{t-1}}{\Delta t}) + r_s$ 
 $r_e = 4.0 - r_a + r_s$ |
| MO-Hopper | $r_r$: forward running 
 $r_j$: vertical jumping | $r_r = 1.5 \cdot \frac{x_t - x_{t-1}}{\Delta t} + r_s - r_a$ 
 $r_j = 12 \cdot \frac{h_t - h_{init}}{\Delta t} + r_s - r_a$ |
| MO-Hopper-3obj | $r_r$: forward running 
 $r_j$: vertical jumping 
 $r_e$: energy efficiency | $r_r = 1.5 \cdot \frac{x_t - x_{t-1}}{\Delta t} + r_s$ 
 $r_j = 12 \cdot \frac{h_t - h_{init}}{\Delta t} + r_s$ 
 $r_e = 4.0 - r_a + r_s$ |
| MO-Swimmer | $r_v$: forward speed 
 $r_e$: energy efficiency | $r_v = \frac{x_t - x_{t-1}}{\Delta t}$ 
 $r_e = 0.3 - 0.15 \cdot r_a$ |
| MO-Walker2d | $r_v$: forward speed 
 $r_e$: energy efficiency | $r_v = \frac{x_t - x_{t-1}}{\Delta t} + r_s$ 
 $r_e = 4.0 - r_a + r_s$ |

## E.2 Behavioral Policy Quality

Each dataset in D4MORL is collected using:

- **Expert policy**: Selected from a large PGMORL[33]-trained ensemble to match the target preference closely.

- **Amateur policy**: **Amateur policy:** Perturbed version of the expert policy, where actions have a certain probability of being stochastic. For most environments, stochastic actions are generated by scaling the expert action by a random factor sampled from $\text{Unif}(0.35, 1.65)$ (65%), while the expert action is retained otherwise (35%). In the MO-Swimmer environment, stochastic actions (35%) are instead uniformly sampled from the action space to better approximate amateur-level performance.

The target preferences used during data collection are uniformly sampled from the full $(n-1)$-dimensional simplex. This promotes diverse trade-offs across objectives and ensures that the dataset covers a wide range of possible preferences. All preference vectors $\omega \in \mathbb{R}^n$ are normalized to satisfy $\omega_i \geq 0$ and $\sum_i \omega_i = 1$.

### E.3 Reward and State Normalization

We introduce how rewards and states are normalized in the D4MORL benchmark [31].

**Reward normalization.** The reward values are normalized for each objective to the $[0, 1]$ range using min-max normalization:
$$r^{\text{norm}} = \frac{r - r_{\min}}{r_{\max} - r_{\min}},$$
where $r_{\min}, r_{\max}$ are computed empirically from the offline dataset for each objective dimension.

**State normalization.** All state vectors are standardized using environment-specific statistics provided by the D4MORL benchmark. Specifically, the raw state $s$ is normalized as:
$$s^{\text{norm}} = \frac{s - \mu}{\sigma},$$
where $\mu$ and $\sigma$ denote the per-dimension mean and standard deviation of the state distribution, computed from the offline dataset.

## F  Implementation Details

We use the Soft-$\chi^2$ divergence as the regularization function $f$ in FairDICE. This function is defined piecewise as:
$$f(x) = \begin{cases} x \log x - x + 1 & \text{if } x < 1 \\ \frac{1}{2}(x-1)^2 & \text{otherwise} \end{cases}$$
This divergence combines the smooth behavior of KL divergence near $x = 0$ with the quadratic growth of the standard $\chi^2$ divergence for larger $x$.

Although the definition of the convex conjugate $f^*(y)$ may suggest a bi-level optimization, once $f(x)$ is specified, both $f^*(y)$ and $(f')^{-1}(x)$ can be obtained in closed form. For the chosen Soft-$\chi^2$ divergence, we have:
$$f^*(y) = \begin{cases} e^y - 1, & y < 0, \\ \frac{1}{2}y^2 + y, & \text{otherwise,} \end{cases} \qquad (f')^{-1}(x) = \begin{cases} e^x, & x < 0, \\ 1 + x, & \text{otherwise.} \end{cases}$$

Therefore, the final FairDICE objective can be computed directly without solving any inner maximization loop.

The FairDICE algorithm, summarized in Algorithm 1, alternates between optimizing the dual variables $(\nu, \mu)$ and updating the policy $\pi$ via weighted behavior cloning. Both the policy $\pi$ and critic $\nu$ networks are implemented as multilayer perceptrons, parameterized by $\psi$ and $\theta$, respectively. The scalar parameters $\mu$ are updated to maximize the desired social welfare function, and we fix $\alpha = 1$ to correspond to the Nash social welfare objective. The initial state distribution $p_0$ is estimated from the offline dataset.

Table 3 provides a summary of our default hyperparameters. The policy and value networks are constructed with three hidden layers, each containing 768 units. Optimization is performed using the Adam optimizer with a learning rate of $3 \times 10^{-4}$ and a discount factor of $\gamma = 0.99$. To study the effect of the regularization coefficient $\beta$—which governs the trade-off between distributional robustness and optimization stability—we conduct a hyperparameter sweep over $\beta \in \{1.0, 0.1, 0.01, 0.001, 0.0001\}$. Our code is available at: `https://github.com/ku-dmlab/FairDICE.git`.

**Algorithm 1** FairDICE

**Input:** Offline dataset $D$, initial state distribution $p_0$, policy $\pi_\theta$, dual parameters $\nu_\psi$, $\mu_i$, divergence regulrarization parameter $\beta$, concave scalarization function $u_i$.

**Output:** Welfare-maximizing policy $\pi_\theta^*$

1: Initialize all parameters
2: **while** not converged **do**
3:     Update $\nu_\psi$, $\mu$ to minimize:

$$\mathcal{L}_{\nu_\psi,\mu} = \mathbb{E}_{s\sim p_0}[(1-\gamma)\nu_\psi(s)] + \mathbb{E}_{(s,a)\sim D}\left[\beta f_0^*\left(\frac{e_{\nu_\psi,\mu}(s,a)}{\beta}\right)\right] + \sum_i u_i^*(-\mu_i)$$

4:     Compute optimal weights:

$$w_{\nu_\psi,\mu}^*(s,a) = \max\left(0, (f')^{-1}\left(\frac{e_{\nu_\psi,\mu}(s,a)}{\beta}\right)\right)$$

5:     Update policy $\pi_\theta$ via weighted behavior cloning:

$$\mathcal{L}_\theta = -\mathbb{E}_{(s,a)\sim D}\left[w_{\nu_\psi,\mu}^*(s,a)\cdot\log\pi_\theta(a\mid s)\right]$$

6: **end while**

Table 3: Implementation Details for MO Environments

| Hyperparameter | Value |
|---|---|
| $\beta$ | $\{1.0, 0.1, 0.01, 0.001, 0.0001\}$ |
| Hidden dim of $\nu_\psi$ and $\pi_\theta$ | 768 (512 for MO-Ant) |
| n_layer of $\nu_\psi$ and $\pi_\theta$ | 3 (4 for MO-Hopper-3obj) |
| Learning rate | $3 \times 10^{-4}$ |
| $\gamma$ (discount factor) | 0.99 |
| Optimizer | Adam |

Values in parentheses indicate environment-specific overrides.

## G   Experiments Compute Resources

All experiments were conducted on a single machine equipped with an Intel® Xeon® Gold 6330 CPU (256GB RAM) and an NVIDIA RTX 3090 GPU. Training a single FairDICE policy on each D4MORL task required approximately 10 to 20 minutes on average. During training, GPU memory usage remained below 20GB.

## H   Robustness of FairDICE to Limited Data Quality and Coverage in Offline Multi-Objective RL

As offline RL methods are often sensitive to the quality and coverage of the dataset, we provide empirical evidence that FairDICE exhibits a degree of robustness to suboptimal data quality and limited coverage. Regarding data quality, we evaluated FairDICE on both the expert and amateur datasets from the D4MORL benchmark, and it consistently achieved high Nash Social Welfare across both settings. To further assess robustness to limited data coverage, we conducted additional experiments where we filtered out trajectories whose preference weights lie near the center of the simplex. Specifically, we removed trajectories in which all preference weights fall between 0.4 and 0.6. This filtering removes data points likely to represent balanced or fair trade-offs, resulting in a more challenging offline dataset.

Table 4: Comparison of Nash Social Welfare before and after trajectory filtering across different environments and dataset qualities.

| Environment | Dataset Quality | % Traj. Removed | NSW (Full) | NSW (Filtered) |
|---|---|---|---|---|
| MO-Swimmer-v2 | Expert | 24.0% | $11.597 \pm 0.091$ | $11.489 \pm 0.192$ |
| MO-Swimmer-v2 | Amateur | 24.0% | $11.820 \pm 0.005$ | $11.819 \pm 0.001$ |
| MO-Walker2d-v2 | Expert | 34.9% | $11.534 \pm 0.039$ | $9.562 \pm 1.161$ |
| MO-Walker2d-v2 | Amateur | 35.0% | $11.396 \pm 0.291$ | $11.339 \pm 0.016$ |
| MO-Ant-v2 | Expert | 43.1% | $11.535 \pm 0.018$ | $11.320 \pm 0.332$ |
| MO-Ant-v2 | Amateur | 43.2% | $11.509 \pm 0.049$ | $11.384 \pm 0.027$ |
| MO-HalfCheetah-v2 | Expert | 43.6% | $11.828 \pm 0.017$ | $11.714 \pm 0.035$ |
| MO-HalfCheetah-v2 | Amateur | 43.9% | $11.994 \pm 0.146$ | $11.709 \pm 0.074$ |
| MO-Hopper-v2 | Expert | 67.3% | $11.058 \pm 0.395$ | $11.157 \pm 0.014$ |
| MO-Hopper-v2 | Amateur | 67.7% | $11.570 \pm 0.003$ | $11.548 \pm 0.003$ |

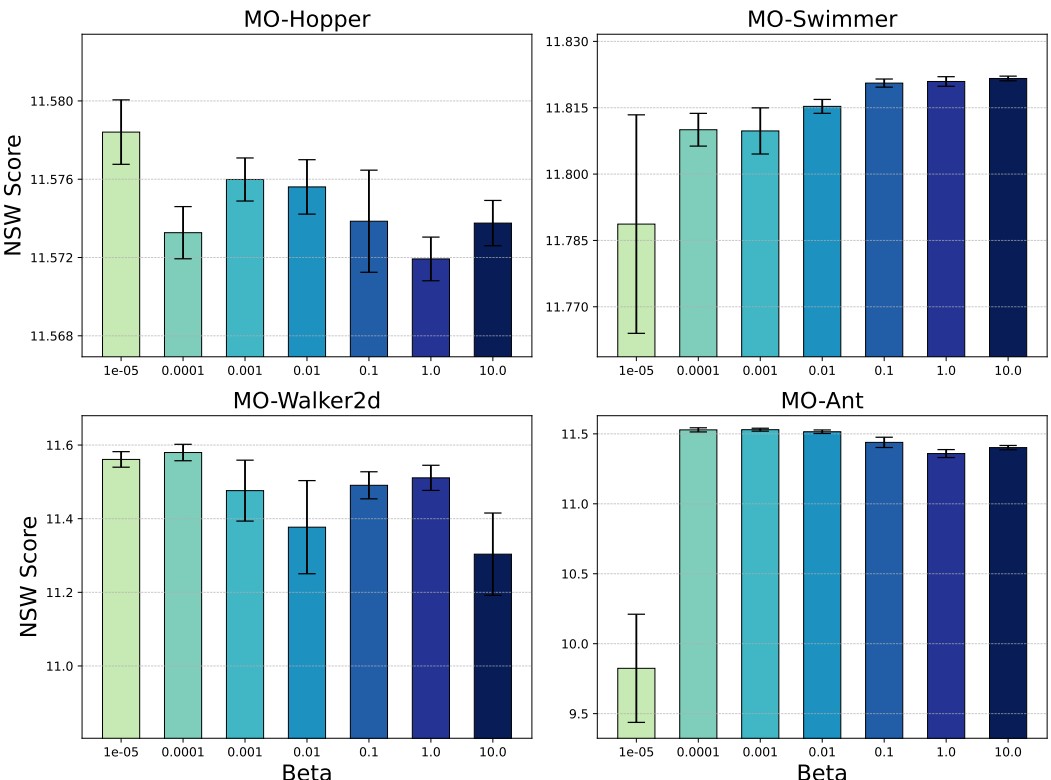

Figure 8: Performance of FairDICE with varying $\beta$ values on amateur datasets in D4MORL. Results are averaged over 10 seeds and 10 episodes, with error bars denoting ±1 standard errors.

The results are shown in 4. Nash Social Welfare (Full) refers to performance on the original dataset without any filtering, while Nash Social Welfare (Filtered) reports performance after removing the trajectories. Each result shows the average Nash Social Welfare (NSW) over 5 seeds. FairDICE continued to perform reliably, demonstrating its ability to optimize fairness-driven objectives even under biased or sparse data conditions.

# I   Impact of f-divergence on FairDICE

In this section, we investigate how varying $\beta$, which controls the strength of regularization, affects FairDICE's performance. Figure 8 reports Nash social welfare (NSW) performance for $\beta \in \{10.0, 1.0, 0.1, 0.01, 0.001, 0.0001, 0.00001\}$. We show that the trade-off between NSW and distributional shift observed in FairDICE in the finite domain (Figure 2) generally extends to the continuous domain. NSW performance typically improves as $\beta$ decreases, reflecting a stronger emphasis on maximizing NSW and reduced reliance on the dataset distribution. While FairDICE

displays strong NSW across a wide range of $\beta$, excessive distributional shift at very small $\beta$ can degrade its practical performance.

An exception is observed in the MO-Swimmer environment, where NSW consistently decreases as $\beta$ decreases. This is likely because the MO-Swimmer dataset already contains trajectories with high NSW, making further deviation harmful. This is supported by Figure 4, which shows that although trajectories were generated using different preference weights in existing preference-based baselines, they consistently achieve high NSW.

