# OpenReview forum: "FairDICE: Fairness-Driven Offline Multi-Objective Reinforcement Learning"
_NeurIPS.cc/2025/Conference — NeurIPS 2025 poster_

### Official Review · Reviewer_SCn5 · 2025-06-30

**Clarity:** 3
**Significance:** 3
**Originality:** 2
**Rating:** 5
**Confidence:** 4

**Summary:**

This paper makes the following contributions:

1. It proposes FairDICE, a novel method for addressing the offline multi-objective reinforcement learning (MORL) problem.

2. It provides an analytical perspective on FairDICE, showing its equivalence to linear MORL under specific weighting schemes.

3. It demonstrates the effectiveness of the proposed method through extensive experiments, supports the theoretical analysis, and includes ablation studies on hyperparameters.

**Questions:**

1. I am curious about the behavior of $\mu$ during training. Does it continue to evolve throughout the entire training process, or does it typically converge quickly in the early stages and remain relatively stable afterward?
If the latter is true, one could potentially view the algorithm as an optimizer for $\mu$, which could then be combined with other algorithms that perform well in linear MORL settings—instead of relying on DICE as the policy solver. Would this be a viable direction?

**Ethical Concerns:**

["NO or VERY MINOR ethics concerns only"]

**Final Justification:**

I appreciate the authors’ detailed responses and am satisfied with the clarifications provided. I have also read the rebuttals to other reviewers and found no points that significantly changed my overall evaluation. This is a solid and worthwhile submission.

That said, I believe the work is not particularly strong in terms of novelty, and there is still room for improvement in the design and completeness of the experiments, including ablations, as also noted by other reviewers. Therefore, I do not consider an increase in score to be warranted and will retain my current rating.

**Limitations:**

Yes.

**Quality:**

4

**Strengths And Weaknesses:**

### Strengths

1. The writing is clear and fluent, with a well-balanced level of detail in both the main text and the appendix. The theoretical derivations are easy to follow.

2. The proposed idea is natural yet novel: building on the DICE framework, the authors incorporate Lagrangian methods to adaptively handle nonlinear multi-objective reinforcement learning (MORL) settings.

3. The paper includes detailed ablation studies and analyses regarding the hyperparameters involved in the method.

4. The experimental setup is comprehensive, covering both continuous and discrete action space environments.

---

### Weaknesses

1. To the best of my knowledge, there are no major flaws. The following points are minor and do not significantly affect the overall quality of the paper.

2. A small suggestion regarding notation: the paper refers to the Bellman flow constraint using “(2)”, which sometimes appears at the end of equations. This can be misleading, as it may be mistaken for a subscript or label—particularly in the definitions of P2, P3, and P2-reg. While careful readers will not be confused, using a clearer reference format (e.g., avoiding placing “(2)” at the end of expressions) could improve the reading experience.

3. Typos:
   - Line 158: The position of $\sum_{\bar{s}, \bar{a}}$ appears to be incorrect.
   - In the loss expression below line 159, a summation symbol seems to be missing before the final term $u_i(k_i)$.

---

> ### Author Rebuttal · Authors · 2025-07-31
>
> Dear Reviewer SCn5,
>
> We sincerely thank you for your close reading and constructive suggestions. We hope the responses below address your comments.
>
> >**[Q1: Behavior and Role of $\mu$ During Training and Its Potential Use with Other MORL Algorithms]**
>
> In our experiments, we observe that the dual variable $\mu$, which implicitly represents the preference weights across objectives, tends to converge relatively early in the training process (e.g., stabilizing around 20k steps out of total 100k training steps). This suggests that FairDICE quickly identifies the correct trade-off priorities by the fairness objective.
>
> However, while your suggestion of two-stage approach (using FairDICE to find an optimal $\mu^{\\ast}$ and then plugging it into another high-performing linear MORL algorithm) sounds appealing, this direction is non-trivial due to the tight coupling between the policy optimization and the preference weight optimization in our framework.
>
> The key point is that the equivalence we established in Section 5.2 holds **specifically for the regularized setting**. The optimal weights $\mu^{\\ast}$ are optimal only for the regularized problem. Without regularization, as we show in Appendix B, the optimal policy for Fair MORL (which can be stochastic) can be fundamentally different from the one found by linear MORL (often deterministic), even if they use the same weights. Therefore, one cannot simply take the $\mu^{\\ast}$ from our **regularized** FairDICE and plug it into a different family of offline RL algorithms (like CQL or IQL) and expect it to remain optimal. Those algorithms employ entirely different regularization principles to handle distribution shift. The optimality of $\mu^{\\ast}$ is tied to the specific DICE-style optimization landscape.
>
> >**[W2, W3: Notation Clarity for Bellman Flow Constraint and Minor Typographical Corrections]**
>
> We sincerely appreciate your suggestion regarding the use of “(2)” to refer to the Bellman flow constraint. To avoid potential confusion with subscripts or equation components—especially in the definitions of P2, P3, and P2-reg—we will adopt a clearer notation. Specifically, we will use $BF(d) = 0$ to denote the Bellman flow constraint after it is formally defined in the paper.
>
> We will also fix the following typos in the updated manuscript:
>
>   * Line 158: Fixing the summation expression by rewriting as $\sum_{s}\nu(s)\sum_{\bar{s},\bar{a}}T(s|\bar{s},\bar{a})d(\bar{s},\bar{a})$.
>
>   * Line 159: Revising $\\sum_{i}\\mu_{i}k_{i}+u_{i}(k_{i})$ to $\\sum_{i}(\\mu_{i}k_{i}+u_{i}(k_{i}))$ to ensure correctness.

---

> > ### Comment · Reviewer_SCn5 · 2025-08-04
> >
> > Thank you for detailed response. I am satisfied with the clarifications provided. I have also read the authors’ rebuttals to other reviewers, and I did not find any points that significantly changed my overall evaluation of the paper. This is a solid and worthwhile submission.
> >
> > That said, I personally believe the work is not particularly strong in terms of novelty, and there is still room for improvement in the design and completeness of the experiments, including ablations—as also noted by other reviewers. Therefore, I do not believe an increase in score is warranted, and I will retain my current rating.

---

### Official Review · Reviewer_s6Th · 2025-07-02

**Clarity:** 3
**Significance:** 2
**Originality:** 3
**Rating:** 5
**Confidence:** 3

**Summary:**

This paper investigates an important problem of fairness in multi-objective reinforcement learning (MORL) in the offline RL setting. Instead of using the standard linear scalarization function to optimize the expected sum of rewards, the authors propose to directly optimize nonlinear social welfare functions, which they claim better capture fairness goals such as Nash Social Welfare and $\alpha$-fairness. To this end, they develop FairDICE, an offline MORL method that optimizes a nonlinear welfare objective while leveraging distribution correction estimation to handle distributional shift. The proposed method is evaluated across several domains and demonstrates promising results compared to existing baselines.

**Questions:**

See above and also below for more questions:

1. How would FairDICE perform under different welfare functions, such as the generalized Gini or WOWA?

2. Could the proposed fairness mechanism be extended to other offline RL methods, such as CQL, IQL, or BC?

3. Offline RL methods are often sensitive to the quality and coverage of the dataset. How robust is FairDICE when data coverage is poor or biased, especially in real-world problems where multiple and possibly conflicting objectives are present?

[1] Siddique, Umer, Paul Weng, and Matthieu Zimmer. "Learning fair policies in multi-objective (deep) reinforcement learning with average and discounted rewards." International Conference on Machine Learning. PMLR, 2020.

[2] Yu, Guanbao, Umer Siddique, and Paul Weng. "Fair deep reinforcement learning with preferential treatment." ECAI 2023. IOS Press, 2023. 2922-2929.

[3] Kumar, Aviral, et al. "Conservative q-learning for offline reinforcement learning." Advances in neural information processing systems 33 (2020): 1179-1191.

[4] Kostrikov, Ilya, Ashvin Nair, and Sergey Levine. "Offline reinforcement learning with implicit q-learning." arXiv preprint arXiv:2110.06169 (2021).

**Ethical Concerns:**

["NO or VERY MINOR ethics concerns only"]

**Final Justification:**

The authors have addressed all of my questions and provided clear clarifications to my concerns. Specifically, my questions regarding the DICE-RL framework and the choice of the welfare function were thoroughly addressed. Based on these clarifications, I am increasing my scores accordingly.

**Limitations:**

The limitations are adequately addressed by the authors.

**Paper Formatting Concerns:**

I did not notice any major formatting issues in this paper.

**Quality:**

3

**Strengths And Weaknesses:**

Strengths:
* The paper is well written and generally easy to follow.

* The use of a nonlinear welfare function to address fairness in the offline MORL setting is an interesting and important contribution. To my knowledge, this is indeed the first work to formalize and solve the fair optimization problem in offline MORL.

* I particularly like how the authors frame the fair MORL problem as a convex optimization problem and use a concave welfare function for policy learning. This is not only theoretically sound but also practically vert important and relevant.
* The experimental results are also strong, comprehensive and considered baselines are strong and very relevant.

Overall, I like this work. It addresses an underexplored but practically important problem. However, I have some concerns and questions, which I list below.

Weaknesses:
* The current work focuses on $\alpha$-fairness, which has some well-known limitations. For example, if some utilities are negative, the Nash Social Welfare (which is equivalent to the geometric mean) becomes infeasible because it requires all utilities to be positive. This same issue can affect the general $\alpha$-fairness for other values of $\alpha$. This restricts practical applicability in real-world scenarios where some objectives may yield negative returns. I recommend that the authors also explore and if possibly include other general-purpose welfare functions, such as the Generalized Gini welfare function or Weighted Ordered Weighted Average (WOWA) as it has been considered in online RL settings [1-2].

* Although the paper claims to be the first method to tackle fairness in offline RL, the proposed method relies heavily on the DICE-RL framework. It would be useful to explain why DICE-RL was chosen instead of other SOTA offline RL baselines such as CQL [3], IQL [4], or BC. For example, can the proposed method be integrated with these methods? This point is especially relevant given that similar extensions have been demonstrated in the online RL setting [1].

* Another potential drawback of FairDICE is its dependence on the hyperparameter $\beta$. As shown in the results, the performance appears quite sensitive to the choice of $\beta$ which introduces additional tuning complexity on top of standard RL hyperparameters. More discussion on choosing this particular hyperparameter would be beneficial for the reader.

---

> ### Author Rebuttal · Authors · 2025-07-31
>
> Dear Reviewer s6Th,
>
> Thank you for taking the time to review our work. We greatly value your feedback and aim to address your concerns thoroughly in the following response.
>
> >**[W1,Q1:Limitations of Alpha-Fairness with Negative Returns and Recommendation to Consider Alternative Welfare Functions like Gini or WOWA]**
>
> We agree with the reviewer’s observation that the welfare functions used in our paper can become infeasible when negative returns are present, particularly functions like $\log(x)$, which are undefined for non-positive values. A practical workaround for handling negative returns is to define the utility function piecewise. For the non-positive domain, we can use a concave function that handles negative inputs, and attach it to the original $\log(x)$ utility function. For example, we can define the utility for $x<1$ as a quadratic segment, say $-0.5(x−2)^2 + 0.5$, and attach it smoothly to $\log(x)$ for $x \geq 1$. By doing so, we can ensure the function remains continuous, differentiable, and concave while being well-defined for all real-valued returns. Although this modifies the original Nash Social Welfare (NSW) objective slightly, it preserves its core fairness-promoting properties and can provide meaningful and robust solutions in practice.
>
> Alternatively, one can consider general-purpose welfare functions such as the Generalized Gini Function (GGF) and the Weighted Ordered Weighted Averaging (WOWA), which naturally handle negative returns. However, there are two structural differences that make optimizing rank-based objectives such as GGF or WOWA directly within the FairDICE framework nontrivial.
>
> First, the primary challenge with rank-based objectives like GGF or WOWA is the instability and optimization difficulties this introduces. These methods define the welfare function based on the **sorted order** of the objective returns. Since any off-policy estimation (OPE) of returns from a fixed dataset is inherently noisy, even small estimation errors can cause the predicted ranks of two closely-valued objectives to flip.  Furthermore, this rank-switching can make the overall objective function **non-smooth or even discontinuous** with respect to the policy being optimized. This can lead to unstable learning signals, making it challenging to optimize particularly in the **offline** setting. It is likely for this very reason that the works for GGF and WOWA mentioned by the reviewer, do so exclusively in the **online** setting.
>
> Furthermore, this practical challenge points to a difference in how these frameworks encode fairness. GGF and WOWA rely on explicit, discrete sorting to determine an objective's importance. In contrast, FairDICE uses a continuous and implicit mechanism: preference is not determined by rank, but is encoded directly into the smooth curvature of the concave utility function. This yields naturally smooth and stable prioritization based on an objective's current return level, which can lead to more stable optimization.
>
> We appreciate the reviewer’s suggestion and agree that exploring alternative welfare formulations, particularly those that accommodate negative returns, is a promising direction for extending FairDICE to broader real-world scenarios. Such formulations could enable more stable and robust optimization of welfare in environments with high variance or unbounded return scales.
>
> >**[W2, Q2: Justification for Using DICE-RL Over Other Offline RL Methods Like CQL or IQL, and Potential for Integration]**
>
> Our choice of DICE-RL framework was driven by the unique structural requirements of Fair MORL objective, which are not easily addressed by value-based methods like CQL or IQL.
>
> The key advantage of DICE-RL is its formulation as a convex optimization problem over the space of **stationary distributions** (d). This structure is well-suited for our objective, $\sum_{i} u_i( J_i )$ where the expected returns $J_i$ are linear functions of the distribution $d$: $J_i = E_d [r_i]$. This allows us to use convex optimization techniques (i.e., introducing slack variables) to handle the concave utility function $u_i$ in a principled way, circumventing the difficult nested expectation problem.
>
> In contrast, adapting value-based methods to non-linear fair objectives is fundamentally challenging. A naive approach might be to learn separate Q-functions $Q_i$ for each objective, and then derive a greedy policy. However, unlike standard linear scalarization, the standard policy improvement guarantee breaks down for non-linear objectives due to the fact that $\sum_{i} u_i( J_i( \pi ) )] \neq E_{\pi}[ \sum_{i} u_i( J_i(s,a) )]$. Because these two quantities are not equivalent, there is no guarantee that optimizing the latter will improve the former (our true goal). Therefore, naively applying value-based methods would fail to learn truly optimal fair policy. It remains unclear how to optimize a fairness-driven policy when the scalarization is non-linear.
>
> In light of such a challenge, investigating how to adapt SOTA offline RL algorithms like CQL and IQL to the fairness-aware MORL setting is a highly promising direction for future work. It has strong potential to improve the practical performance and scalability of fair MORL methods in complex environments.
>
> >**[W3: Concern About Sensitivity to Hyperparameter $\beta$ and Need for Practical Guidance on Its Selection]**
>
> We agree that the hyperparameter $\beta$ plays a critical role in balancing the trade-off between welfare maximization and distributional shift, and as such, it can influence the final performance of FairDICE.
>
> However, our empirical results across both discrete and continuous domains indicate that FairDICE maintains robust performance for sufficiently small values of $\beta$. To support this claim, we refer to the results in both domains. In the discrete setting (Random MOMDP, Figure 2), FairDICE consistently achieves high Nash Social Welfare across broad range of relatively small $\beta$ values (e.g., from 0.001 to 1). This suggests that finding a suitable $\beta$ is not a matter of delicate tuning. Similarly, in the continuous domain, as shown in Appendix H, FairDICE exhibits stable performance across a broad range of $\beta$ values.
>
> While it is true that offline RL algorithms can be sensitive to hyperparameter choices, our findings suggest that FairDICE is relatively robust in this regard. We will revise the paper to clarify this point and provide additional guidance on selecting $\beta$ in practice.
>
> >**[Q3: Robustness of FairDICE to Limited Data Quality and Coverage in Offline Multi-Objective RL]**
>
> As offline RL methods are often sensitive to the quality and coverage of the dataset, we provide empirical evidence that FairDICE exhibits a degree of robustness to suboptimal data quality and limited coverage.
>
> Regarding data quality, as shown in Section 7, we evaluated FairDICE on both the expert and amateur datasets from the D4MORL benchmark [1], and it consistently achieved high Nash Social Welfare across both settings.
>
> To further assess robustness to limited data coverage, we conducted additional experiments where we filtered out trajectories whose preference weights lie near the center of the simplex. Specifically, we removed trajectories in which all preference weights fall between 0.4 and 0.6. This filtering removes data points likely to represent balanced or fair trade-offs, resulting in a more challenging offline dataset.
>
> | Environment   	| Dataset Quality | % Traj. Removed | Nash Social Welfare (Full)| Nash Social Welfare (Filtered) |
> | ----------------- | --------------- | --------------: | -----------------------: | ---------------------------: |
> | MO-Swimmer-v2 	| Expert      	|       	24.0% |       	11.597 ± 0.091 |           	11.489 ± 0.192 |
> | MO-Swimmer-v2 	| Amateur     	|       	24.0% |       	11.820 ± 0.005 |           	11.819 ± 0.001 |
> | MO-Walker2d-v2	| Expert      	|       	34.9% |       	11.534 ± 0.039 |            	9.562 ± 1.161 |
> | MO-Walker2d-v2	| Amateur     	|       	35.0% |       	11.396 ± 0.291 |           	11.339 ± 0.016 |
> | MO-Ant-v2 	    | Expert      	|       	43.1% |       	11.535 ± 0.018 |           	11.320 ± 0.332 |
> | MO-Ant-v2     	| Amateur     	|       	43.2% |       	11.509 ± 0.049 |           	11.384 ± 0.027 |
> | MO-HalfCheetah-v2 | Expert      	|           43.6% |       	11.828 ± 0.017 |           	11.714 ± 0.035 |
> | MO-HalfCheetah-v2 | Amateur     	|       	43.9% |       	11.994 ± 0.146 |           	11.709 ± 0.074 |
> | MO-Hopper-v2  	| Expert      	|       	67.3% |   	    11.058 ± 0.395 |           	11.157 ± 0.014 |
> | MO-Hopper-v2  	| Amateur     	|       	67.7% |       	11.570 ± 0.003 |           	11.548 ± 0.003 |
>
> The results are shown in the table above. Nash Social Welfare (Full) refers to performance on the original dataset without any filtering, while Nash Social Welfare (Filtered) reports performance after removing the trajectories. Each result shows the average Nash Social Welfare (NSW) over 5 seeds. FairDICE continued to perform reliably, demonstrating its ability to optimize fairness-driven objectives even under biased or sparse data conditions.
>
> We will incorporate these additional robustness results into the updated version of the manuscript to better highlight FairDICE’s robustness.
>
> **References:**
>
> \[1] Baiting Zhu, Meihua Dang, and Aditya Grover. *Scaling pareto-efficient decision making via offline multi-objective rl*. In The Eleventh International Conference on Learning Representations, 2023.

---

> > ### Comment · Reviewer_s6Th · 2025-08-05
> >
> > Thank you for the detailed response. The authors have addressed all of my questions and effectively clarified my concerns. I appreciate the effort and clarity in the rebuttal. Good job, I am satisfied with the response and willing to increase my scores accordingly.

---

### Official Review · Reviewer_YvDZ · 2025-07-02

**Clarity:** 3
**Significance:** 4
**Originality:** 3
**Rating:** 5
**Confidence:** 3

**Summary:**

This paper presents FairDICE, a method to learn fairness-aware policies in offline multi-objective RL. Improving on the limitations of existing MORL techniques that rely on linear combinations of objectives, this approach allows for learning concave welfare functions which require non-linear scalarization. The authors cast fair MORL as a convex optimization problem, and extend ideas from DICE-RL, using regularized optimization to avoid distribution shift from the offline dataset while still learning welfare-oriented policies. The authors derive the optimization using a primal-dual formulation. They also show equivalence of the proposed algorithm to the regularized linear MORL.
Finally, they perform experiments on discrete and continuous domains, and show FairDICE is able to learn fair or utilitarian policies, based on the alpha-fairness measure with different weights. They also discuss the evolution of these methods when random noise is added and when they are trained on one metric but measured on another. Finally, the authors show the policies generated by FairDICE lie on the pareto front when compared to other baselines with exhaustive search.

**Questions:**

Q1. Why is the non-linearity an issue? The bellman equation can still hold as long as the non-linearity is within a single time-step. Is there a time-dependence for the non-linearity?

Q2. Why is the equivalence between fair MORL and regularized  linear MORL important?

Q3. Why is FairMORL a single point in Figure 5? Is it possible to elicit different tradeoffs? If yes, where do they lie in this figure, and why is this not compared?

**Ethical Concerns:**

["NO or VERY MINOR ethics concerns only"]

**Final Justification:**

The authors sufficiently clarified my doubts about the paper. I will maintain my score.

**Limitations:**

yes

**Quality:**

3

**Strengths And Weaknesses:**

Strengths:
1. The paper introduces fair MORL in the offline setting, providing a new and performant method for learning policies with non-linear scalarizations as required by many welfare functions. This is important for settings where balancing between objectives is key.
2. The authors also extensively evaluate the method and show its strength compared to reasonable baselines and analyze the behavior in different regimes.
3. The theoretical formulation for the method gives it credence.

Weaknesses:
1. The paper assumes that u is a concave function (line 113), without defining in what sense this concavity is applied. Is it concave over all objectives? Is it concave over time? While readers can figure this out, clearly stating this and defining $u$ concretely will help.
2. It is not clear why the equivalence between the regularized linear and fair MORL methods is important. The authors should discuss the significance, as significant space is spent outlining this in the paper.
3. In Figure 5, FairDICE is a single point, and often lies close to MORvSP. It seems MORvSP is similarly performant and able to capture a wider range of pareto-optimal solutions. If it is easy to train FairDICE with specific welfare functions, why are there not multiple points for FairDICE in this figure? It signals a lack of control rather than a strength (e.g. why is the tradeoff selected by FairDICE any good?)

---

> ### Author Rebuttal · Authors · 2025-07-31
>
> Dear Reviewer YvDZ,
>
> We sincerely thank you for your detailed comments and insightful suggestions. We hope that our responses below provide the necessary clarification and help resolve the points you raised.
>
> >**[W1: Clarification Needed on Domain and Interpretation of Concavity Assumption for Utility Function $u$]**
>
> To clarify, the concavity we refer to applies individually to each objective's return component, not over time steps. That is, the total objective is of the form $\sum_{i}u_{i}(R_{i})$, where each $u_{i}$ is a concave function applied to the return of the $i$-th objective.
>
> We use concave functions to promote fairness across objectives. This approach is motivated by the principle of diminishing marginal utility, which implies that an identical gain in return provides more utility to underperforming objectives than to well-performing ones. As a result, the optimization naturally prioritizes weaker objectives, learning to more balanced and equitable outcomes.
>
> This design choice is also supported by prior literature on fairness in multi-objective optimization, where per-objective concave functions or fairness-aware metrics (e.g., Nash Social Welfare) have been commonly used to balance outcomes [1, 2, 3]. We will revise the paper to clearly explain both the interpretation and the motivation behind this formulation.
>
> >**[W2,Q2: Clarification Needed on the Importance of the Equivalence Between Regularized Linear MORL and Fair MORL]**
>
> The equivalence we establish serves a crucial **theoretical and practical purpose**: it demonstrates precisely why FairDICE offers a more principled and efficient approach compared to naive approaches.
>
> **Theoretically**, the equivalence proves that for a given fairness objective (e.g. Nash Social welfare), there exists a **single, specific** set of preference weights that a regularized linear MORL agent would need to find the truly optimal fair policy.
>
> **Practically**, however, finding these "magic" weights with a linear MORL method poses a major challenge. Since these optimal weights are unknown beforehand, one must perform brute-force grid search, a process that becomes computationally infeasible due to the curse of dimensionality as the number of objectives increases.
>
> This is where the significance of FairDICE becomes clear. Instead of requiring this inefficient, extensive search, FairDICE treats the preference weights as optimization variables, directly learning the optimal weights as part of its end-to-end training process. In essence, it provides a principled way to find the optimal fair policy in a single run, without any weight sweeping.
>
> Although this motivation is briefly mentioned in the final paragraph of Section 5, we agree that a clearer and more explicit explanation would help readers better understand this advantage. We will revise the text accordingly in the final version.
>
> >**[W3,Q3: Why is FairMORL a single point in Figure 5? Is it possible to elicit different tradeoffs? If yes, where do they lie in this figure, and why is this not compared?]**
>
> We would like to clarify that the primary goal of FairDICE differs fundamentally from that of linear MORL methods like MORvSP. Linear MORL approaches aim to find  a wide range of Pareto-optimal solutions, leaving the final decision of which trade-off is "best" to the user. In contrast, FairDICE, along with other non-linear fair MORL methods [1, 2, 3], is designed to  directly find a single, principled solution that is optimal under a pre-defined, meaningful fairness criterion (e.g., Nash Social Welfare). Therefore, the single point in Figure 5 is a deliberate feature of our design, not a limitation.
>
> In Figure 5, FairDICE appears as a single point because it optimizes a fixed nonlinear scalarization function: Nash Social Welfare (NSW), defined by $u_i(x) = \log x$. However, FairDICE allows for eliciting different trade-offs by changing the underlying utility function. For example, the widely used α-fairness formulation allows interpolation across a spectrum of fairness criteria:
>
> $
> u_i(x) =
> \\begin{cases}
> \\frac{x^{1-\\alpha}}{1 - \\alpha} & \\text{if } \\alpha \\ne 1 \\\\
> \\log(x) & \\text{if } \\alpha = 1
> \\end{cases}
> $
>
> Smaller values of $\alpha$ (e.g., $\alpha \to 0$) emphasize efficiency and resemble utilitarian welfare, whereas larger values (e.g., $\alpha \to \infty$) prioritize equality, approaching max-min fairness. In addition, a weighted Nash objective, such as $u_i(x) = w_i \log x$, can be used to reflect asymmetric priority across objectives. Each of these alternative formulations would yield a different point on or near the Pareto front.
>
> While we did not compare multiple trade-offs in this figure, we agree that such an analysis could provide additional insight. We will clarify this design choice in the paper and discuss the potential to explore alternative fairness trade-offs as part of future work.
>
> **References:**
>
> \[1] Zimeng Fan, Nianli Peng, Muhang Tian, and Brandon Fain. *Welfare and fairness in multi-objective reinforcement learning*. arXiv preprint arXiv:2212.01382, 2022.
>
> \[2] Giseung Park, Woohyeon Byeon, Seongmin Kim, Elad Havakuk, Amir Leshem, and Youngchul Sung. *The max-min formulation of multi-objective reinforcement learning: From theory to a model-free algorithm*. arXiv preprint arXiv:2406.07826, 2024.
>
> \[3] Mridul Agarwal, Vaneet Aggarwal, and Tian Lan. *Multi-objective reinforcement learning with non-linear scalarization*. In Proceedings of the 21st International Conference on Autonomous Agents and Multiagent Systems, pages 9–17, 2022.

---

> > ### Comment · Reviewer_YvDZ · 2025-08-05
> >
> > Thanks for the detailed response. The authors have addressed most of my questions. If possible, I'd like the authors to also comment on Q1, regarding non-linearity. Even in the face of non-linear functions, why are traditional Q-learning methods off the table? The Bellman-style updates should be able to handle arbitrary rewards given state-action pairs.

---

> > > ### Author Response · Authors · 2025-08-07
> > >
> > > Dear Reviewer YvDZ,
> > >
> > > >**[Q1. Why is the non-linearity an issue? The bellman equation can still hold as long as the non-linearity is within a single time-step. Is there a time-dependence for the non-linearity?]**
> > >
> > > We agree that if the non-linearity were applied *within each time step*, Bellman-style updates could in principle still be used. However, in our setting, the non-linearity is **not** applied to immediate rewards, but to the **total expected return** for each objective:
> > >
> > > $$
> > > \sum_i u_i\left( \mathbb{E}_\pi \left[ \sum^\infty _{t=0}\gamma^t r_i(s_t, a_t) \right] \right),
> > > $$
> > >
> > > This structure is essential for fair MORL objectives like Nash Social Welfare. However, applying non-linear function $u_i$ (e.g., log) to the **time-aggregated return** breaks the recursive structure fundamental to the Bellman equation.
> > >
> > > As a result, the problem is **no longer time-decomposable**. It becomes impossible to define a local, per-timestep Temporal Difference (TD) target, rendering standard Q-learning and other value-based TD methods unsuitable.
> > >
> > > This is precisely the challenge our work addresses. FairDICE circumvents this by reformulating the problem in the space of stationary distributions and using slack variables. This allows us to solve this non-time-decomposable problem in a principled manner, which is beyond the scope of traditional Q-learning.

---

> > > > ### Comment · Reviewer_YvDZ · 2025-08-09
> > > >
> > > > Thanks for the clarification. I have no follow-ups. I will maintain my score

---

### Official Review · Reviewer_Rez3 · 2025-07-02

**Clarity:** 3
**Significance:** 2
**Originality:** 2
**Rating:** 4
**Confidence:** 4

**Summary:**

This paper studies the problem of multi-objective offline reinforcement learning with fairness-oriented goals. Fairness requires a nonlinear and non-additive combination of objectives which makes the standard linear scalarization method ineffective. This work extends DICE-RL framework to the multi-objective setting which adopts the regularized linear programming formulation of RL. Based on this framework, they derived a practical sample-based objective FairDICE for fairness-oriented goals. They also build the relation between FairDICE and the standard linear formulation by showing FairDICE implicitly optimizes the linear weights. Finally, they conducted experiments to show the effectiveness of their method.

**Questions:**

1. How is the FairDICE objective (4) optimized practically?
2. If the setting is online, can we design a practical algorithm based on the dual formulation (3)?

**Ethical Concerns:**

["NO or VERY MINOR ethics concerns only"]

**Final Justification:**

I am recommending an accept. It is a good and clear work on fair offline multi-agent RL. I didn't give a score of 5 considering the technique of introducing slack variables is a standard trick in optimization area and the regularized offline RL framework also builds on DICE-RL which makes me feel the work has a moderate not high impact.

**Limitations:**

Yes.

**Paper Formatting Concerns:**

No.

**Quality:**

3

**Strengths And Weaknesses:**

Strengths:
1. This work studies the setting of fairness-oriented multi-objective offline reinforcement learning which is relatively under-studied.
2. The experiments are comprehensive including both discrete and continuous domains.
3. It is an interesting result that optimizing the regularized fairness objective is implicitly optimizing the preference weights of all objectives.

Weaknesses:
1. The main contribution of this work is the FairDICE objective for multi-objective offline RL problem. However, formulating fair MORL as a convex optimization (P1) and converting it to the dual formulation (P2) is a standard derivation. It is not clear what specific challenges are and how they are solved. And with (P2), the derivation of the final FairDICE objective is to incorporate the existing DICE-RL objective to take the distribution shift into account. Based on these facts, I think the innovation of this work is a little bit limited.
2. Different from the objective of DICE-RL, the FairDICE objective contains convex conjugate functions which makes it a bi-level  optimization problem, especially the term $f^*(e_{\nu,\mu}(s,a))$. In the paper, there is no explanation on how the objective is optimized practically.

---

> ### Author Rebuttal · Authors · 2025-07-31
>
> Dear Reviewer Rez3,
>
> Thank you for your thoughtful and constructive feedback. We appreciate your careful review and hope that the following response addresses your questions and alleviates your concerns.
>
> >**[W1: Clarification on Novelty Beyond Standard Convex Reformulation and DICE-RL]**
>
> To the best of our knowledge, introducing slack variables to derive an equivalent convex optimization problem is novel in the context of reinforcement learning. In typical formulations of Fair MORL, the objective takes the form of a concave utility function applied to an expected return, as in problem (P1). A direct approach to solving (P1) is challenging because the expectation is nested inside the non-linear (concave) function, which leads to **biased** sample-based estimation.
>
> To address this challenge, we introduce slack variables to move the term $\sum_{s,a} d(s,a) r_i(s,a)$ outside the concave utility function $u_i$ (to linear constraints)​, resulting in a reformulated problem (P2). This crucial step enables us to develop a sample-based algorithm, FairDICE, which **circumvents the bias issue** incurred by taking expectation inside the concave function.
>
> Although the derivation borrows elements from DICE-RL to account for distribution shift, simply applying DICE to a multi-objective setting is non-trivial—especially when fairness and optimality must be preserved across conflicting objectives. Our contribution lies in identifying how to retain these properties within a convex dual structure, which, to our knowledge, had not been done before in this context.
>
> >**[W2,Q1: Lack of Explanation on Practical Optimization of FairDICE’s Bi-Level Objective with Convex Conjugate Term]**
>
> The use of a maximization in the definition of the convex conjugate may give the impression of a bi-level optimization. However, **once the function $f(x)$ is specified, its conjugate $f^{*}(y)$ can be derived in closed form**. For example, in our implementation of FairDICE (Appendix F), we use the soft-$\chi^2$ divergence as the regularization:
>
> $
> f(x) =
> \\begin{cases}
> x \\log x - x + 1, & x < 1 \\\\
> 0.5(x - 1)^2, & x \\geq 1
> \\end{cases}
> \\quad \\text{and} \\quad u(x) = \\log x.
> $
>
> The corresponding conjugate functions and $(f')^{-1}(x)$ are given analytically by:
>
> $
> f^{\\ast}(y) =
> \\begin{cases}
> e^y - 1, & y < 0 \\\\
> 0.5y^2 + y, & \\text{otherwise}
> \\end{cases}
> \\quad \\text{and} \\quad u^*(y) = 1 + \\log y.
> $
>
> $
> (f')^{-1}(x) =
> \\begin{cases}
> e^x, & x < 0 \\\\
> 1+x, & \\text{otherwise}
> \\end{cases}
> $
>
> Therefore, the **final objective can be computed directly and does not require solving an inner optimization loop**.
>
> The practical optimization procedure is provided in Algorithm 1 in Appendix F (p. 25). To improve clarity and accessibility, we will revise the main text to include a concrete example of the regularization function $f$, the utility function $u$, and their convex conjugates, and clearly reference the relevant section of the Appendix where the full algorithm is described.
>
> >**[Q2: If the setting is online, can we design a practical algorithm based on the dual formulation (3)?]**
>
> Equation (3) by itself is not directly suitable for constructing a practical online algorithm, for two main reasons. First, the variable $d(s,a)$ does not generally correspond to the stationary distribution of any policy unless it satisfies the Bellman flow constraints. Second, even if we were to compute expectations with respect to $d(s,a)$, the $F_d(s)$ term depends on transition probabilities for arbitrary $(s,a)$, which are typically unknown in standard model-free settings (both online and offline). These challenges limit the direct applicability of (3) and motivate the subsequent derivations in our paper that transform it into a practical, sample-based objective.

---

> > ### Comment · Reviewer_Rez3 · 2025-08-05
> >
> > Thanks the authors for their careful clarifications. I am keeping my score, and lean towards acceptance. I didn't give a score of 5 considering the technique of introducing slack variables is a standard trick in optimization area and the regularized offline RL framework also builds on DICE-RL which makes me feel the work has a moderate not high impact.

---

### Comment · Area_Chair_Ax7G · 2025-08-05

Dear Reviewers

Thanks for participating in discussion. The author-reviewer discussion period is extended to Aug. 8, 11:59PM AoE.

If you have not yet, I kindly ask you to

- Read the author rebuttal and other reviews.
- Post your response to the author rebuttal as soon as possible.
- Engage in discussion with the authors and other reviewers where necessary.

It is not permitted by the conference policy that a reviewer submits mandatory acknowledgement without a single reply sentence to author rebuttal.

If you have already engaged in author-reviewer discussion, I sincerely thank you. If not, I kindly ask you to engage in discussion.

Thank you very much for your help.

Best regard,

AC

---

### Comment · Area_Chair_Ax7G · 2025-08-08

Dear Reviewers

Thank you again for participating in reveiwer-author discussion.

The reviewer-author discussion is coming to an end soon. If you did not submit mandatory acknowledgement yet, please submit the ack, final score and justification by the end of the reviewer-author discussion period.

Thanks.

Area Chair

---

### Note · Authors · 2025-08-14

Dear Area Chair and Reviewers,

We want to thank you again for your time, consideration, and constructive engagement during this period. We have been able to significantly improve the manuscript with your valuable input.

We have addressed all concerns in detail in our individual responses. To briefly summarize our core contribution in light of the novelty discussions: our main innovation lies in providing the first tractable, principled solution to the non-time-decomposable problem of offline fair MORL, a challenge previously unaddressed by standard RL methods.

We will incorporate all promised revisions into the final version of the paper. We thank you once more for your careful review and feedback.

---

### Decision · Program_Chairs · 2025-09-17

**Decision:**

Accept (poster)

**Comment:**

This paper considers fairness-aware policy learning in off-line multi-objective reinforcement learning, and proposes a nice solution to this problem based on the DICE framework. Theoretical analysis and numerical validations are sound.  AC recommends the acceptance of the paper. Please include the additional results obtained during the rebuttal period in the final version.